# AdaSVD: Adaptive Singular Value Decomposition for Large Language Models

## Abstract

Large language models (LLMs) have achieved remarkable success in natural language processing (NLP) tasks, yet their substantial memory requirements present significant challenges for deployment on resource-constrained devices. Singular Value Decomposition (SVD) has emerged as a promising compression technique for LLMs, offering considerable reductions in memory overhead. However, existing SVD-based methods often struggle to effectively mitigate the errors introduced by SVD truncation, leading to a noticeable performance gap when compared to the original models. Furthermore, applying a uniform compression ratio across all transformer layers fails to account for the varying importance of different layers. To address these challenges, we propose `AdaSVD`, an adaptive SVD-based LLM compression approach. Specifically, `AdaSVD` introduces **adaComp**, which adaptively compensates for SVD truncation errors by alternately updating the singular matrices $\mathcal{U}$ and $\mathcal{V}^{\top}$. Additionally, `AdaSVD` introduces **adaCR**, which adaptively assigns layer-specific compression ratios based on the relative importance of each layer. Extensive experiments across multiple LLM/VLM families demonstrate that `AdaSVD` consistently outperforms state-of-the-art (SOTA) SVD-based methods, achieving superior performance with significantly reduced memory requirements. We will release all the code and models of `AdaSVD`.

## 1 Introduction

Recently, large language models (LLMs) based on the Transformer architecture (Vaswani, 2017) have shown remarkable potential across a wide range of natural language processing (NLP) tasks. However, their success is largely driven by their massive scale, with models such as the LLaMA family (Touvron et al., 2023a) and the Open Pre-trained Transformer (OPT) series (Zhang et al., 2022) containing up to 70B and 66B parameters, respectively. The substantial memory requirements of these models present significant challenges for deploying them on mobile devices. Consequently, the widespread adoption of LLMs remains limited by their immense resource demands (Wan et al., 2023; Wang et al., 2024; Zhou et al., 2024).

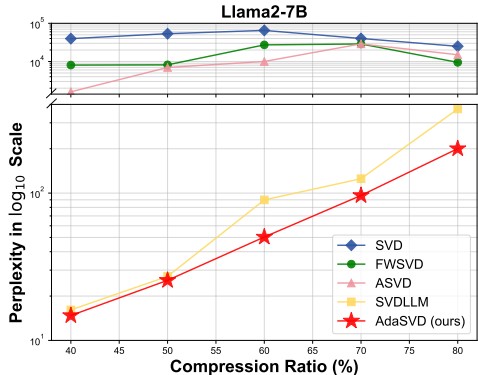

Figure 1: Comparison between vanilla SVD, FWSVD (Hsu et al., 2022a), ASVD (Yuan et al., 2024), SVD-LLM (Wang et al., 2025), and our `AdaSVD` on WikiText2.

Recent research on large language model (LLM) compression has explored various techniques, including weight quantization (Lin et al., 2024; Frantar et al., 2023), network pruning (Sun et al., 2024; Frantar & Alistarh, 2023), low-rank factorization (Wang et al., 2025; Zhang et al., 2024; Yuan et al., 2024), and knowledge distillation (Zhong et al., 2024; Gu et al., 2024). Among these methods, low-rank factorization using Singular Value Decomposition (SVD) (Hsu et al., 2022a; Yuan et al., 2024; Wang et al., 2025) stands out as a powerful approach for reducing both model size and computational cost. SVD achieves this by decomposing large weight matrices into smaller, low-rank components while preserving model performance. Since LLMs are often memory-bound during

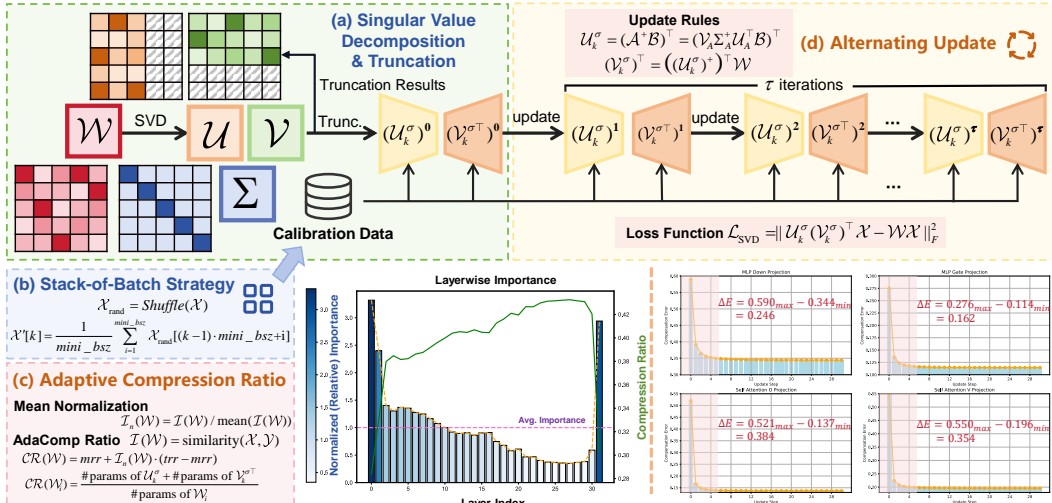

Figure 2: Overview of the proposed `AdaSVD` method: (a) SVD decomposition and truncation for linear layer weights; (b) Stack-of-batch strategy for efficient use of calibration data under limited GPU memory; (c) Adaptive compression ratio assignment (**adaCR**) based on layer-wise importance; (d) Adaptive compensation (**adaComp**) through alternating updates of $\mathcal{U}$ and $\mathcal{V}^{\top}$.

inference (Dao et al., 2022; Dao, 2024), SVD compression can effectively accelerate model inference by reducing the memory requirements, even when applied solely to the weights. This approach does not require specialized hardware or custom operators, unlike weight quantization, making SVD more versatile across different platforms. Additionally, SVD is orthogonal to other compression techniques (Wang et al., 2025), allowing it to be combined with methods like weight quantization or network pruning for even greater efficiency.

Recent advancements in SVD-based LLM compression, including FWSVD (Hsu et al., 2022a), ASVD (Yuan et al., 2024), and SVD-LLM (Wang et al., 2025), have significantly improved the low-rank factorization approach, enhancing the overall effectiveness of SVD compression. For example, FWSVD introduces Fisher information to prioritize the importance of parameters, while ASVD accounts for the impact of activation distribution on compression error. SVD-LLM establishes a relationship between singular values and compression loss through the data whitening techniques. While these methods have led to notable improvements in SVD compression, they still face challenges when applied at high compression ratios.

To bridge the performance gap between compressed and original models at both low and high compression ratios, we revisit SOTA solutions for LLM compression using SVD decomposition. Our analysis highlights two key observations: **First,** low-rank weight compensation after truncating the smallest singular vectors has been largely overlooked or insufficiently explored in prior methods. When truncating parts of the matrices $\mathcal{U}$ and $\mathcal{V}^{\top}$, the remaining components should be adjusted accordingly to minimize the SVD compression error. **Second,** previous methods typically apply a uniform compression ratio across all the transformer layers, not taking into account their relative importance. Thus, an importance-aware approach is needed to assign appropriate compression ratios.

To tackle the challenges outlined above, we propose `AdaSVD`, an adaptive SVD-based LLM compression method. **First,** AdaSVD proposes **adaComp**, an adaptive compensation technique designed to adjust the weights of $\mathcal{U}$ and $\mathcal{V}^{\top}$ after SVD truncation. By alternately updating the matrices $\mathcal{U}$ and $\mathcal{V}^{\top}$, **adaComp** effectively reduces compression errors in a stable and efficient manner. To optimize the use of calibration data with limited GPU memory, we also introduce a stack-of-batch technique when applying **adaComp**. **Second,** AdaSVD proposes **adaCR**, a method that assigns adaptive compression ratios to different layers based on their importance. With fixed target compression ratio, this strategy significantly improves performance compared to using a uniform compression ratio across all layers.

Our key contributions are summarized as follows:

- We propose **adaComp**, a novel adaptive compensation method for SVD truncation. By alternately updating $\mathcal{U}$ and $\mathcal{V}^{\top}$ and employing the stack-of-batch technique, we effectively and stably minimize compression error.

---

**Algorithm 1** Pseudocode of `AdaSVD`

---

1: **Inputs:** LLM $\mathcal{M}$, Calib Data $\mathcal{C}$, Bucket Size $M$, Target Retention Ratio $trr$, Min Retention Ratio $mrr$, Update Iteration $k$
2: **Outputs:** Updated Model $\mathcal{M}'$ by `AdaSVD`
3: **procedure** ADASVD($\mathcal{M}, \mathcal{C}, trr, mrr, k$)
4:      $\mathcal{X} \leftarrow$ GET_CALIB($\mathcal{C}$)                  ▷ Randomly collect samples as calibration data
5:      $\mathcal{X}'[1], \mathcal{X}'[2], ..., \mathcal{X}'[M] \leftarrow$ SOB($\mathcal{X}, M$)       ▷ Shuffle samples and utilize SOB strategy
6:      Set$_{\mathcal{S}} \leftarrow$ WHITENING($\mathcal{M}, \mathcal{X}'$), Set$_{\mathcal{SVD}} \leftarrow \emptyset$, Set$_{\mathcal{W}} \leftarrow \mathcal{M}$
7:      Set$_{\mathcal{CR}} \leftarrow$ LAYER_CR($\mathcal{M}, \mathcal{X}', trr, mrr$)           ▷ Measure layerwise importance
8:      **for** layer $i$ **in** language model $\mathcal{M}$ **do**
9:          $\mathcal{W}_i \leftarrow$ Set$_{\mathcal{W}}(i)$, $\mathcal{S}_i \leftarrow$ Set$_{\mathcal{S}}(\mathcal{W}_i)$    ▷ Extract the whitening matrix of current weight $\mathcal{W}_i$
10:         $\mathcal{U}_i, \Sigma_i, \mathcal{V}_i \leftarrow$ SVD($\mathcal{W}_i \mathcal{S}_i$)              ▷ Apply Singular Value Decomposition
11:         $\Sigma' \leftarrow$ TRUNC($\Sigma_i$), $(\mathcal{U}_i', \mathcal{V}_i') \leftarrow$ TRUNC_UV($\mathcal{U}, \mathcal{V}, \Sigma'$)     ▷ Truncate with adaptive ratio
12:         Set$_{\mathcal{SVD}} \leftarrow (\mathcal{U}_i', \mathcal{V}_i') \cup$ Set$_{\mathcal{SVD}}$
13:      **end for**
14:      $\mathcal{M}' \leftarrow$ ADA_UPDATE($\mathcal{M}, \mathcal{X}'$, SET$_{\mathcal{SVD}}, k$)     ▷ Alternate update $\mathcal{U}_i', \mathcal{V}_i'$ for $k$ iterations
15:      **return** $\mathcal{M}'$
16: **end procedure**

---

- We propose **adaCR**, an adaptive compression ratio method that assigns layer-specific compression ratios according to their relative importance in LLMs. This importance-aware approach outperforms the previously used uniform compression ratio method.

- Extensive experiments on LLMs/VLMs demonstrate that our method, `AdaSVD`, significantly outperforms the previous SVD-based LLM compression method, SVD-LLM, effectively narrowing the performance gap between compressed and original models.

## 2 RELATED WORKS

### 2.1 LLM COMPRESSION TECHNIQUES

Recent advancements in model compression techniques have significantly enhanced the efficiency of deploying LLMs while maintaining their performance. Widely explored approaches include weight quantization (Frantar et al., 2023; Lin et al., 2024), network pruning (Frantar & Alistarh, 2023; Ma et al., 2023; Yang et al., 2024; Gromov et al., 2025; Ashkboos et al., 2024), and hybrid methods (Dong et al., 2025). In unstructured pruning, SparseGPT (Frantar & Alistarh, 2023) prunes weights based on their importance, as determined by the Hessian matrix. However, it faces challenges in achieving optimal speedup, particularly due to hardware compatibility issues. Structured pruning methods, in contrast, are more hardware-friendly. LLM-Pruner (Ma et al., 2023) selectively removes non-critical coupled structures using gradient information. LaCo (Yang et al., 2024) introduces a layer-wise pruning strategy, where subsequent layers collapse into preceding ones. Gromov et al. (2025) explores the effectiveness of basic layer-pruning techniques combined with parameter-efficient fine-tuning (PEFT). Additionally, SliceGPT (Ashkboos et al., 2024) has pioneered post-training sparsification, emphasizing the importance of layer removal order for optimal performance. Quantization techniques offer another significant avenue for compression. GPTQ (Frantar et al., 2023) applies layer-wise quantization and reduces quantization errors through second-order error compensation. AWQ (Lin et al., 2024) introduces activation-aware weight quantization, employing a scale transformation between weights and activations. Moreover, BiLLM (Huang et al., 2024) and ARB-LLM (Li et al., 2025) achieve further compression to 1-bit while maintaining remarkable performance. However, many of these compression techniques face challenges related to hardware compatibility, often requiring custom CUDA kernels (Dong et al., 2025) to enable real-time inference speedup.

### 2.2 SVD-BASED LLM COMPRESSION

Singular Value Decomposition (SVD) is a widely used technique for reducing matrix size by approximating a matrix with two smaller, low-rank matrices (Golub et al., 1987). Although SVD-based methods have demonstrated potential in compressing LLMs, their full capabilities remain underexplored. Standard SVD typically focuses on compressing the original weight matrix without considering the significance of individual parameters, which can lead to considerable compression

errors. To address this, Hsu et al. (2022b) introduced FWSVD, which incorporates Fisher information to weight the importance of parameters. However, this method requires complex gradient calculations, making it resource-intensive. Another limitation of standard SVD is the impact of activation distribution on compression errors. To mitigate this, Yuan et al. (2024) proposed ASVD, which scales the weight matrix with a diagonal matrix that accounts for the influence of input channels on the weights. Subsequently, Wang et al. (2025) introduced SVD-LLM, which establishes a connection between singular values and compression loss. This work demonstrates that truncating the smallest singular values after data whitening effectively minimizes compression loss. Despite these advancements, existing methods still exhibit significant accuracy loss at higher compression ratios and lack a comprehensive approach for compensating compressed weights after SVD truncation. Furthermore, most methods apply a uniform compression ratio across all transformer layers, overlooking the varying importance of them. `AdaSVD` seeks to address these limitations by an adaptive compensation method (**adaComp**) and an importance-aware adaptive compression ratio method (**adaCR**), respectively.

## 3 METHOD

**Overview.** As illustrated in Figure 2, `AdaSVD` integrates adaptive compensation for SVD truncation (**adaComp**) with an adaptive importance-aware compression ratio method (**adaCR**). In Section 3.1, we first describe how **adaComp** compensates for SVD truncation. Then, in Section 3.2, we detail how **adaCR** determines the compression ratio based on layer importance. The pseudocode of `AdaSVD` is shown in Algorithm 1, and pseudocodes of **adaComp** and **adaCR** are provided in supplementary file.

### 3.1 ADAPTIVE COMPENSATION FOR SVD TRUNCATION

SVD compression first applies SVD decomposition for matrix $\mathcal{W}$, and then truncates the smallest singular values:

$$\mathcal{W} = \mathcal{U}\Sigma\mathcal{V}^\top \approx \mathcal{U}_k\Sigma_k\mathcal{V}_k^\top = \widehat{\mathcal{W}}, \quad (1)$$

where $\Sigma_k$ indicates the retaining top-k largest singular values, $\mathcal{U}_k$ and $\mathcal{V}_k^\top$ represent the corresponding retaining singular vectors. Moreover, the diagonal matrix $\Sigma_k$ can be further absorbed into $\mathcal{U}_k$ and $\mathcal{V}_k^\top$ by

$$\mathcal{U}_k^\sigma = \mathcal{U}_k\Sigma_k^{\frac{1}{2}}, \ \mathcal{V}_k^\sigma = \mathcal{V}_k\Sigma_k^{\frac{1}{2}}, \quad (2)$$

$$\widehat{\mathcal{W}} = \mathcal{U}_k\Sigma_k\mathcal{V}_k^\top = \mathcal{U}_k^\sigma(\mathcal{V}_k^\sigma)^\top. \quad (3)$$

The truncation of the smallest singular values minimizes the compression error with respect to $\mathcal{W}$, ensuring that $||\mathcal{U}_k^\sigma(\mathcal{V}_k^\sigma)^\top - \mathcal{W}||_F^2$ is minimized, which we refer to as the vanilla SVD method. However, this approach does not fully account for the practical effects of $\mathcal{X}$. To address this limitation, we introduce a more application-relevant metric for the SVD compression error, defined as follows:

$$\mathcal{L}_{\text{SVD}} = ||\widehat{\mathcal{W}}\mathcal{X} - \mathcal{W}\mathcal{X}||_F^2$$
$$= ||\mathcal{U}_k^\sigma(\mathcal{V}_k^\sigma)^\top\mathcal{X} - \mathcal{W}\mathcal{X}||_F^2. \quad (4)$$

Previous works (Hsu et al., 2022b; Yuan et al., 2024; Wang et al., 2025) have made significant efforts to minimize $\mathcal{L}_{\text{SVD}}$. However, some of them involve complex and time-consuming

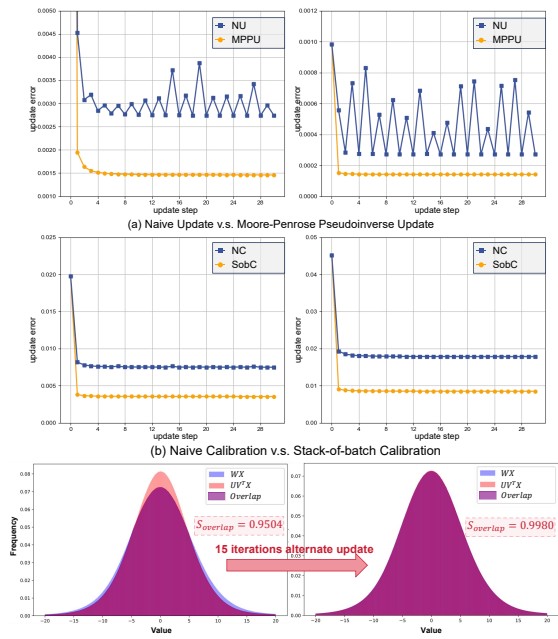

Figure 3: Adaptive compensation for SVD truncation (**adaComp**). (a) Comparison between naive (NU) and Moore-Penrose pseudoinverse update (MPPU). (b) Comparison between naive (NC) and stack-of-batch calibration strategy (SobC). (c) Distribution comparison before and after applying **adaComp**.

preprocessing steps. Furthermore, they still face substantial challenges in effectively mitigating large errors that arise under high compression ratios, particularly when truncating 60% or more parameters.

To compensate for the error attributed to SVD truncation, we need to optimize the following objective:

$$\mathcal{U}_k^\sigma, \mathcal{V}_k^{\sigma\top} = \arg\min_{\mathcal{U}_k^\sigma, \mathcal{V}_k^{\sigma\top}} ||\mathcal{U}_k^\sigma\mathcal{V}_k^{\sigma\top}\mathcal{X} - \mathcal{W}\mathcal{X}||_F^2. \quad (5)$$

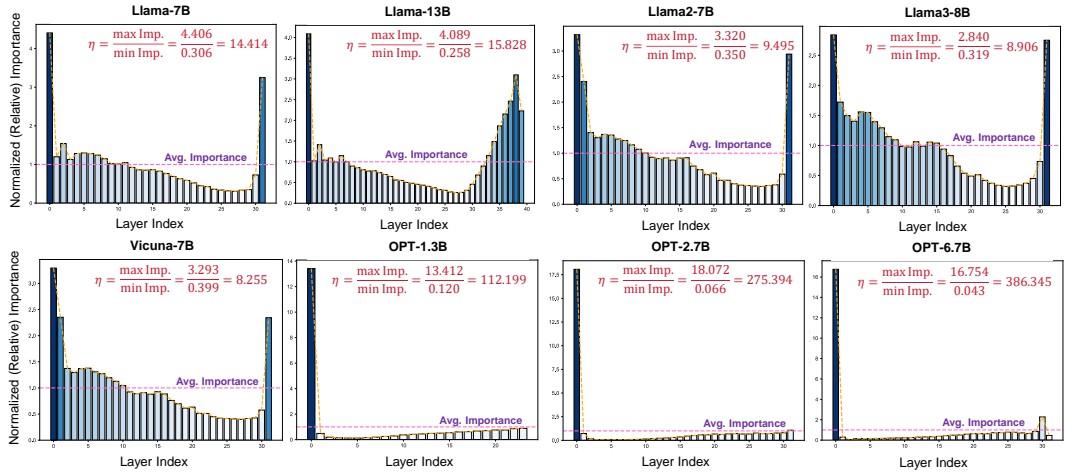

Figure 4: Layer-wise relative importance of different LLMs. The importance across different layers varies significantly, and the first layer always weight most importance. More layer-wise importance visualization can be found in the supplementary file.

A straightforward approach is to compute the partial derivatives of the SVD compression objective with respect to $\mathcal{U}_k^\sigma$ and $\mathcal{V}_k^{\sigma\top}$, resulting in the following expressions (additional details can be found in the supplementary file):

$$\frac{\partial \mathcal{L}_{\text{SVD}}}{\partial \mathcal{U}_k^\sigma} = 0 \implies \mathcal{U}_k^\sigma = \mathcal{W}\mathcal{X}\mathcal{X}^\top \mathcal{V}_k^\sigma ((\mathcal{V}_k^\sigma)^\top \mathcal{X}\mathcal{X}^\top \mathcal{V}_k^\sigma)^{-1}, \tag{6}$$

$$\frac{\partial \mathcal{L}_{\text{SVD}}}{\partial \mathcal{V}_k^{\sigma\top}} = 0 \implies \mathcal{V}_k^{\sigma\top} = ((\mathcal{U}_k^\sigma)^\top \mathcal{U}_k^\sigma)^{-1} (\mathcal{U}_k^\sigma)^\top \mathcal{W}. \tag{7}$$

However, this method involves computing the matrix inverse, which can lead to unstable updates and significant compression errors, as shown in Figure 3 (a). To mitigate the issue of numerical instability, we propose a two-fold strategy to enhance the update quality of $\mathcal{U}_k^\sigma$ and $\mathcal{V}_k^{\sigma\top}$.

**First**, the optimization objective for $\mathcal{U}_k^\sigma$ is reformulated as a Least Squares Estimation (LSE) problem, where $\mathcal{V}_k^{\sigma\top}\mathcal{X}$ is treated as the input and $\mathcal{W}\mathcal{X}$ as the output:

$$\mathcal{U}_k^\sigma = \arg\min_{\mathcal{U}_k^\sigma} \|\mathcal{A}(\mathcal{U}_k^\sigma)^\top - \mathcal{B}\|_F^2, \tag{8}$$

where $\mathcal{A} = \mathcal{X}^\top \mathcal{V}_k^\sigma$ and $\mathcal{B} = (\mathcal{W}\mathcal{X})^\top$. Since $\mathcal{A}$ is typically not a square matrix and may not be full rank, we first apply SVD to $\mathcal{A}$ to enhance numerical stability:

$$\mathcal{A} = \mathcal{U}_\mathcal{A} \Sigma_\mathcal{A} \mathcal{V}_\mathcal{A}^\top, \tag{9}$$

and then obtain the solution for $\mathcal{U}_k^\sigma$ by using the Moore-Penrose pseudoinverse (Penrose, 1955) of $\mathcal{A}$:

$$\mathcal{U}_k^\sigma = (\mathcal{A}^+ \mathcal{B})^\top = (\mathcal{V}_\mathcal{A} \Sigma_\mathcal{A}^+ \mathcal{U}_\mathcal{A}^\top \mathcal{B})^\top, \tag{10}$$

where $\Sigma_A^+$ denotes the Moore-Penrose pseudoinverse of $\Sigma_A$:

$$\Sigma_A = \text{diag}(\sigma_1, \sigma_2, \ldots, \sigma_n), \tag{11}$$

$$\Sigma_A^+ = \text{diag}\left(\sigma_1^{-1}\mathbb{1}_{\sigma_1 \neq 0}, \sigma_2^{-1}\mathbb{1}_{\sigma_2 \neq 0}, \ldots, \sigma_n^{-1}\mathbb{1}_{\sigma_n \neq 0}\right). \tag{12}$$

Similarly, we update $\mathcal{V}_k^{\sigma\top}$ by the Moore-Penrose pseudoinverse of $\mathcal{U}_k^\sigma$ to handle numerical instability:

$$\mathcal{V}_k^{\sigma\top} = \arg\min_{\mathcal{V}_k^{\sigma\top}} \|\mathcal{U}_k^\sigma \mathcal{V}_k^{\sigma\top} \mathcal{X} - \mathcal{W}\mathcal{X}\|_F^2 = \left((\mathcal{U}_k^\sigma)^+\right)^\top \mathcal{W}. \tag{13}$$

As shown in Figure 3 (a), by reformulating the optimization objective as an LSE problem and solving for $\mathcal{U}$ and $\mathcal{V}^\top$ using the Moore-Penrose pseudoinverse, we achieve a smooth curve that consistently reduces compression error stably.

**Second**, since the update rule incorporates the calibration data $\mathcal{X}$, ideally, a large volume of $\mathcal{X}$ would yield better results. However, during our experiments, we found that extending $\mathcal{X}$ to just 32 samples on an 80GB GPU is challenging. To address this, we propose a **stack-of-batch** strategy that enables the utilization of more calibration data without increasing memory overhead. Specifically, given $N$ calibration samples and a bucket size $M$ (the maximum number of samples that can fit within the

Table 1: Zero-shot performance comparison of LLaMA2-7B between `AdaSVD` and previous SVD compressed methods under 40% to 60% compression ratios. Evaluation on three language modeling datasets (measured by perplexity ($\downarrow$)) and five common sense reasoning datasets (measured by both individual and average accuracy ($\uparrow$)) demonstrate the effectiveness of `AdaSVD`.

| RATIO | METHOD | WikiText-2↓ | PTB↓ | C4↓ | Mmlu | ARC_e | WinoG. | HellaS. | PIQA | Average↑ |
|-------|--------|-------------|------|-----|------|-------|--------|---------|------|----------|
| 0% | Original | 5.68 | 8.35 | 7.34 | 45.30 | 74.62 | 69.22 | 76.00 | 79.11 | 68.85 |
| 40% | SVD | 39,661.03 | 69,493.00 | 56,954.00 | **26.51** | 26.39 | 48.62 | 25.64 | 52.99 | 36.03 |
| | FWSVD | 8,060.35 | 9,684.10 | 7,955.21 | 25.74 | 26.05 | 50.20 | 25.70 | 52.39 | 36.01 |
| | ASVD | 1,609.32 | 7,319.49 | 1,271.85 | 24.35 | 26.81 | 49.49 | 25.83 | 53.81 | 36.06 |
| | SVD-LLM | 16.11 | 719.44 | 61.95 | 22.97 | 36.99 | 56.04 | 30.49 | 56.96 | 40.69 |
| | AdaSVD | **14.76** (↓8%) | **304.62** (↓58%) | **56.98** (↓8%) | 23.63 | **41.12** | **58.17** | **31.75** | **58.49** | **42.63** |
| 50% | SVD | 53,999.48 | 39,207.00 | 58,558.00 | **25.43** | 25.80 | 47.36 | 25.55 | 52.67 | 35.36 |
| | FWSVD | 8,173.21 | 8,615.71 | 8,024.67 | 24.83 | 25.84 | 48.70 | 25.64 | 52.83 | 35.57 |
| | ASVD | 6,977.57 | 15,539.44 | 4,785.15 | 24.52 | 25.13 | 49.17 | 25.48 | 52.94 | 35.45 |
| | SVD-LLM | 27.19 | 1,772.91 | 129.66 | 23.44 | 31.65 | 51.14 | 28.38 | 54.57 | 37.83 |
| | AdaSVD | **25.58** (↓6%) | **593.14** (↓67%) | **113.84** (↓12%) | 23.24 | **34.18** | **54.06** | **28.88** | **55.50** | **39.17** |
| 60% | SVD | 65,186.67 | 79,164.00 | 70,381.00 | 22.94 | 24.49 | **51.85** | 25.40 | 53.16 | 35.57 |
| | FWSVD | 27,213.30 | 24,962.80 | 47,284.87 | **26.91** | 25.38 | 48.46 | 25.61 | 51.96 | 35.66 |
| | ASVD | 10,003.57 | 15,530.19 | 9,983.83 | 26.89 | 26.68 | 48.86 | 25.76 | 51.80 | 36.00 |
| | SVD-LLM | 89.90 | 2,052.89 | 561.00 | 22.88 | 26.73 | 47.43 | 26.89 | **53.48** | 35.48 |
| | AdaSVD | **50.33** (↓44%) | **1,216.95** (↓41%) | **239.18** (↓57%) | 24.69 | **28.20** | 51.22 | 27.36 | 52.83 | **36.87** |

fixed GPU memory), we randomly sample $mini\_bsz = \lceil \frac{N}{M} \rceil$ samples into one bucket by taking their mean value as follows:

$$\mathcal{X}_{\text{rand}} = Shuffle(\mathcal{X}), \tag{14}$$

$$\mathcal{X}'[k] = \frac{1}{mini\_bsz} \sum_{i=1}^{mini\_bsz} \mathcal{X}_{\text{rand}}[(k-1) \cdot mini\_bsz + i], \tag{15}$$

where $k = 1, 2, \ldots, M$, and cardinality $|\mathcal{X}'| = M$. As shown in Figure 3 (b), integrating the **stack-of-batch** strategy further reduces the compression error.

As shown in Figure 2, to compensate for the error attributed to SVD truncation, we propose an adaptive method to subsequently update $\mathcal{U}_k^\sigma$ and $\mathcal{V}_k^\sigma$ with the above update rules. Moreover, the adaptation of $\mathcal{U}_k^\sigma$ and $\mathcal{V}_k^\sigma$ can be alternatively applied until convergence, where the update sequence over $\tau$ iterations can be expressed as

$$\boxed{(\mathcal{U}_k^\sigma)^1 \to (\mathcal{V}_k^{\sigma\top})^1} \to \boxed{(\mathcal{U}_k^\sigma)^2 \to (\mathcal{V}_k^{\sigma\top})^2} \to \cdots \to \boxed{(\mathcal{U}_k^\sigma)^\tau \to (\mathcal{V}_k^{\sigma\top})^\tau}, \tag{16}$$

where $(\mathcal{U}_k^\sigma)^\tau$ and $(\mathcal{V}_k^{\sigma\top})^\tau$ denote the updated singular matrices after $\tau$-th iteration, respectively, while the region bounded by $\boxed{\phantom{xx}}$ corresponding to one iteration of alternative update. As shown in Figure 3 (c), the gap between the outputs of the compressed and original models narrows after alternative updates. The overlapping area rapidly increases after just a few iterations. More visual comparisons are shown in the supplementary file.

Notably, our adaptive compensation can be integrated with data whitening proposed by Wang et al. (2025) and Liu et al. (2024), further reducing the SVD truncation error.

## 3.2 ADAPTIVE SVD COMPRESSION RATIO

Previous studies on SVD compression typically apply a uniform compression ratio across all transformer layers of LLMs, overlooking the varying importance of different layers. Inspired by Men et al. (2024) and Dumitru et al. (2024), we propose **adaCR**, which adaptively determines the SVD compression ratio for each transformer layer, considering each layer's distinct impact on activations.

The importance of $\mathcal{W}$ can be measured by its impact on the input, which is quantified as the similarity between the input $\mathcal{X}$ and the output $\mathcal{Y}$ after passing through $\mathcal{W}$.

$$\mathcal{Y} = \mathcal{W}\mathcal{X}, \quad \mathcal{I}(\mathcal{W}) = \text{similarity}(\mathcal{X}, \mathcal{Y}), \tag{17}$$

where $\mathcal{I}(\mathcal{W})$ denotes the layer-wise importance of $\mathcal{W}$. The similarity metric used can vary, and for simplicity, we adopt cosine similarity in our method.

Then, we normalize $\mathcal{I}(\mathcal{W})$ through mean centering to obtain the relative importance of $\mathcal{W}$:

$$\mathcal{I}_n(\mathcal{W}) = \mathcal{I}(\mathcal{W})/\text{mean}(\mathcal{I}(\mathcal{W})). \tag{18}$$

After mean normalization, the average importance is 1. A value of $\mathcal{I}_n(\mathcal{W})$ greater than 1 indicates greater importance, while a value lower than 1 indicates lesser importance. The compression ratio of

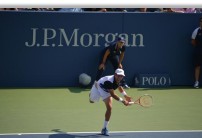

SVD: a man standing on one side of the road and another man sitting on the other side. In this book, it'ans was written by an author who wrote ...

SVD-LLM: Assistants - Asassistant - assasat - as - sasasm - saasam \u2013 Sasmas \u2013 Saas...

AdaSVD: The vehicle is a motorcycle, which is an automobile that is used for transportation. ...

SVD: it's a man ... and he has been injured by an injury that he was not able to play at 20 ...

SVD-LLM: a man who was seen to be an outsider, and he was not able to play at all, but he played with his right hand, which was broken, ...

AdaSVD: It's a person playing tennis on court, and he is wearing white shorts. ... playing his tennis with a tennis racket, ...

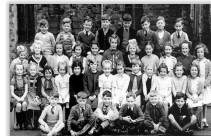

SVD: In this book, it's written by ... John an author of his own books and he is known for his writing ...

SVD-LLM: What'stans was an old name, used by a group of people who were on their way to city...

AdaSVD: It's a photo of children standing together, wearing uniforms... They are sitting on their kneels, it is an old photograph that ...

SVD: ... The mountains are covered with snow, ... difficult for them to climb ... not able to walk through the snow ... trapped in snow holes

SVD-LLM: ... walking on snowy mountains, snow falls down from top of the mountain to bottom ...

AdaSVD: The mountain is covered by snow, ... a group of people skiing and walking on slopes.

Figure 5: We perform image captioning by applying SVD, SVD-LLM (Wang et al., 2025), and AdaSVD to LLaVA-7B on COCO dataset, highlighting correct and wrong captions in different colors.

each layer will be adaptively adjusted based on the relative importance:

$$\mathcal{CR}(\mathcal{W}) = mrr + \mathcal{I}_n(\mathcal{W}) \cdot (trr - mrr), \tag{19}$$

where $mrr$ and $trr$ are the minimum and target retention ratios, respectively. Notably, $\mathcal{CR}(\mathcal{W}) = mrr$ when $\mathcal{I}_n(\mathcal{W}) = 0$, and $\mathcal{CR}(\mathcal{W}) = trr$ when $\mathcal{I}_n(\mathcal{W}) = 1$.

Given the compression ratio for the $i$-th layer by **adaCR**, we truncate the vectors of least singular values from both $\mathcal{U}_k^\sigma$ and $\mathcal{V}_k^{\sigma\top}$ so that

$$\mathcal{CR}(\mathcal{W}_i) = \frac{\#\text{params of } \mathcal{U}_k^\sigma + \#\text{params of } \mathcal{V}_k^{\sigma\top}}{\#\text{params of } \mathcal{W}_i}. \tag{20}$$

As shown in Figure 4, the importance of different layers varies. It can be observed that the first layer always weighs the most importance, suggesting that we should retain more weight on it. For the Llama family, the relative importance curve approximates a bowl shape, highlighting the significance of both the initial and final layers.

## 4 EXPERIMENTS

### 4.1 SETUP

We compare AdaSVD with vanilla SVD and SOTA SVD-based LLM compression methods FWSVD (Hsu et al., 2022b), ASVD (Yuan et al., 2024), and SVD-LLM (Wang et al., 2025).

**Models and Datasets.** To demonstrate the generalizability of our method, we evaluate the performance of AdaSVD and the baselines on four models from three different LLM families, including LLaMA2-7B (Touvron et al., 2023b), OPT-6.7B (Zhang et al., 2022), Mistral-7B (Jiang et al., 2023), and Vicuna-7B (Chiang et al., 2023). We benchmark on eight datasets, including three language modeling datasets (WikiText-2 (Merity et al., 2017), PTB (Marcus et al., 1993), and C4 (Raffel et al., 2020)) and five common-sense reasoning datasets (WinoGrande (Sakaguchi et al., 2019), HellaSwag (Zellers et al., 2019), PIQA (Bisk et al., 2020), ARC-e (Clark et al., 2018), and Mmlu (Hendrycks et al., 2021)). We use the LM-Evaluation-Harness framework (Gao et al., 2023) to evaluate the model performance on these zero-shot Question-Answering (QA) datasets.

Table 2: Perplexity (↓) of four different LLMs – OPT-6.7B, LLaMA 2-7B, Mistral-7B, and Vicuna-7B – under 60% compression ratio on WikiText-2, where AdaSVD shows consistent improvements.

| METHOD | OPT-6.7B | LLaMA2-7B | Mistral-7B | Vicuna-7B |
|---|---|---|---|---|
| SVD | 18,607.24 | 65,186.67 | 30,378.35 | 78,704.50 |
| FWSVD | 8,569.56 | 27,213.30 | 5,481.24 | 8,185.66 |
| ASVD | 10,326.48 | 10,003.57 | 22,705.51 | 20,241.17 |
| SVD-LLM | 92.10 | 89.90 | 72.17 | 64.06 |
| AdaSVD | **86.64** (↓6%) | **50.33** (↓44%) | **67.22** (↓7%) | **56.97** (↓11%) |

**Implementation Details.** To ensure a fair comparison, we followed ASVD (Yuan et al., 2024) and SVD-LLM (Wang et al., 2025) to randomly select 256 samples from WikiText-2 as the calibration data and conduct data whitening before SVD truncation. All experiments are conducted by PyTorch (Paszke et al., 2019) on a single NVIDIA A100-80GB GPU.

### 4.2 MAIN RESULTS

We evaluate the overall performance of AdaSVD from three aspects: **(1)** performance under different compression ratios **(40%, 50%, 60%, 70%, and 80%)**, **(2)** performance on different LLMs. **(3)** per-

Table 3: Perplexity of SVD-compressed LLaMA2-7B, with best results highlighted in ▢.

(a) Effectiveness of Adaptive Compensation

| Method | Tgt. CR | adaComp | WikiText2 ↓ | C4 ↓ |
|--------|---------|---------|-------------|------|
| SVD-LLM | 40% | ✗ | 16.11 | 61.95 |
| AdaSVD | 40% | ✗ | 15.47 | 66.29 |
| AdaSVD | 40% | ✓ | 14.76 | 56.98 |
| SVD-LLM | 50% | ✗ | 27.19 | 129.66 |
| AdaSVD | 50% | ✗ | 30.00 | 166.02 |
| AdaSVD | 50% | ✓ | 25.58 | 113.84 |
| SVD-LLM | 60% | ✗ | 89.90 | 561.00 |
| AdaSVD | 60% | ✗ | 78.82 | 339.31 |
| AdaSVD | 60% | ✓ | 50.33 | 239.18 |

(b) Effectiveness of Adaptive Compression Ratio

| Method | Tgt. CR | CR | WikiText2 ↓ | C4 ↓ |
|--------|---------|-----|-------------|------|
| SVD-LLM | 40% | Const | 16.11 | 61.95 |
| AdaSVD | 40% | Const | 15.38 | 60.43 |
| AdaSVD | 40% | Adapt | 14.76 | 56.98 |
| SVD-LLM | 50% | Const | 27.19 | 129.66 |
| AdaSVD | 50% | Const | 27.33 | 126.85 |
| AdaSVD | 50% | Adapt | 25.58 | 113.84 |
| SVD-LLM | 60% | Const | 89.90 | 561.00 |
| AdaSVD | 60% | Const | 69.46 | 336.90 |
| AdaSVD | 60% | Adapt | 50.33 | 239.18 |

(c) Iteration Number for Adaptive Compression

| Method | Tgt. CR | #Iter | WikiText2 ↓ | C4 ↓ |
|--------|---------|-------|-------------|------|
| SVD-LLM | 40% | - | 16.11 | 61.95 |
| AdaSVD | 40% | 1 | 14.76 | 56.98 |
| AdaSVD | 40% | 3 | 15.47 | 57.28 |
| AdaSVD | 40% | 15 | 15.84 | 57.39 |
| SVD-LLM | 50% | - | 27.19 | 129.66 |
| AdaSVD | 50% | 1 | 25.58 | 113.84 |
| AdaSVD | 50% | 3 | 27.11 | 115.51 |
| AdaSVD | 50% | 15 | 27.45 | 110.35 |
| SVD-LLM | 60% | - | 89.90 | 561.00 |
| AdaSVD | 60% | 1 | 50.33 | 239.18 |
| AdaSVD | 60% | 3 | 64.12 | 301.19 |
| AdaSVD | 60% | 15 | 62.34 | 267.29 |

(d) Minimum Retention Ratio for Adaptive CR

| Method | Tgt. CR | MRR | WikiText2 ↓ | C4 ↓ |
|--------|---------|------|-------------|------|
| SVD-LLM | 40% | - | 16.11 | 61.95 |
| AdaSVD | 40% | 0.40 | 15.01 | 57.17 |
| AdaSVD | 40% | 0.45 | 14.85 | 57.08 |
| AdaSVD | 40% | 0.50 | 14.76 | 56.98 |
| SVD-LLM | 50% | - | 27.19 | 129.66 |
| AdaSVD | 50% | 0.40 | 25.58 | 113.84 |
| AdaSVD | 50% | 0.45 | 26.01 | 117.58 |
| AdaSVD | 50% | 0.50 | 27.33 | 126.85 |
| SVD-LLM | 60% | - | 89.90 | 561.00 |
| AdaSVD | 60% | 0.30 | 50.33 | 239.18 |
| AdaSVD | 60% | 0.35 | 53.17 | 256.66 |
| AdaSVD | 60% | 0.40 | 60.08 | 294.26 |

formance on VLMs. Some performance evaluation results and generated contents by the compressed LLMs are included in the supplementary file to provide a more straightforward comparison.

**Performance under Different Compression Ratios.** First, we evaluate the performance of LLaMA2-7B compressed by vanilla SVD, SVD-LLM (Wang et al., 2025), and AdaSVD under compression ratios ranging from 40% to 80% on all 8 datasets in Table 1. On the three language modeling datasets, AdaSVD consistently outperforms SVD and SVD-LLM in all compression ratios. More importantly, AdaSVD exhibits significant advantages over the baselines under higher compression ratios. These results indicate that AdaSVD is more effective in compressing LLMs for more resource-constrained devices such as smartphones and IoT devices. On the five common sense reasoning datasets, AdaSVD also maintains its edge and performs better than the best-performing baseline on most of the datasets and consistently achieves higher average accuracy across all compression ratios. The results of 70% and 80% compression ratios are provided in supplementary file.

**Performance on Different LLMs.** To demonstrate the generability of AdaSVD across different LLMs, we compare AdaSVD and the baselines on four different models OPT-6.7B, LLaMA2-7B, Vicuna-7B, and Mistral-7B – under 60% compression ratio on WikiText-2. As shown in Table 2, AdaSVD consistently outperforms vanilla SVD, FWSVD, ASVD and SVD-LLM on all LLMs, and exhibits more stable performance across different LLMs, especially compared to vanilla SVD and FWSVD. We reproduce FWSVD, ASVD, and SVD-LLM using their official GitHub repositories. FWSVD and ASVD fail on these LLMs with compression ratios under 60%, whereas SVD-LLM and AdaSVD maintain reasonable perplexity in such cases.

**Performance on Visual Language Models.** Note that our AdaSVD can also be applied to visual language models (VLMs) like LLaVA (Liu et al., 2023). Following Lin et al. (2024), we apply SVD compression to the language part of the VLMs since it dominates the model size. As shown in Figure 5, AdaSVD shows better image captioning results than vanilla SVD and SVD-LLM on COCO dataset (Chen et al., 2015) under 40% compression ratio. More image captioning comparisons with various compression ratios can be found in supplementary file.

### 4.3 ABLATION STUDY

We conduct extensive ablation studies in Table 3 to show the effect of key components in our work.

**Effectiveness of Adaptive Compensation.** To validate the effectiveness of the proposed **adaComp**, we compare the PPL results of Llama2-7B with and without **adaComp** on Wikitest-2 and C4 datasets in Table 3a. Results of 70% and 80% compression ratios can be found in the supplementary file. It can be observed that AdaSVD consistently outperforms SVD-LLM after applying **adaComp**, and the performance gap is more significant under high compression ratios.

**Iteration Number.** To investigate the impact of the number of **adaComp** iterations under different compression ratios, we perform an ablation study with 1, 3, and 15 iterations in Table 3c. The results of 70% and 80% compression ratios are provided in supplementary file. Under lower compression ratios, it is observed that just 1 iteration of **adaComp** already outperforms SVD-LLM. However, increasing the number of iterations may lead to overfitting due to the limited calibration data, resulting in a performance drop. In contrast, under higher compression ratios, additional iterations lead to performance improvements, indicating that AdaSVD is more effective in high compression ratio scenarios where previous methods still struggle. This highlights the importance of balancing the number of iterations with available data to avoid overfitting, especially in low compression scales.

**Effectiveness of Adaptive Compression Ratio.** To validate the effectiveness of our **adaCR**, we compared the results after removing **adaCR** (*i.e.*, using constant compression ratios for all layers) from AdaSVD. As shown in Table 3b, AdaSVD already outperforms SVD-LLM without using **adaCR**, while integrating **adaCR** can further enhance the performance across all compression ratios.

**Minimum Retention Ratio.** The minimum retention ratio ($mrr$) in **adaCR** is also crucial, and we investigate the impact of different $mrr$ values in Table 3d for 40%, 50%, and 60% compression ratios (70% and 80% in supplementary file). It can be observed that $mrr$ remains relatively robust at lower compression ratios (40% and 50%), while contributing more at higher compression ratios (60%).

Table 4: AdaSVD with weight quantization method GPTQ.

| RATIO | METHOD | GPTQ-INT4 | WikiText-2↓ | PTB↓ | C4↓ |
|---|---|---|---|---|---|
| 0% | Original | ✗ | 5.68 | 8.35 | 7.34 |
| 40% | SVD-LLM | ✗ | 16.11 | 719.44 | 61.95 |
| | SVD-LLM | ✓ | 33.56 | 1,887.50 | 184.61 |
| | AdaSVD | ✗ | **14.76** | **304.62** | **56.98** |
| | AdaSVD | ✓ | **22.55** | **844.21** | **106.41** |
| 50% | SVD-LLM | ✗ | 27.19 | 1,772.91 | 129.66 |
| | SVD-LLM | ✓ | 41.70 | 2,335.65 | 291.62 |
| | AdaSVD | ✗ | **25.58** | **593.14** | **113.84** |
| | AdaSVD | ✓ | **37.34** | **1,326.55** | **203.11** |
| 60% | SVD-LLM | ✗ | 89.90 | 2,052.89 | 561.00 |
| | SVD-LLM | ✓ | 119.46 | 3,136.60 | 723.80 |
| | AdaSVD | ✗ | **60.08** | **2,137.28** | **294.26** |
| | AdaSVD | ✓ | **82.08** | **1,705.19** | **379.96** |
| 70% | SVD-LLM | ✗ | 125.16 | 6,139.78 | 677.38 |
| | SVD-LLM | ✓ | 159.53 | 2,115.44 | 848.24 |
| | AdaSVD | ✗ | **107.90** | **5,027.62** | **441.33** |
| | AdaSVD | ✓ | **118.75** | **1,606.94** | **466.64** |
| 80% | SVD-LLM | ✗ | 372.48 | 6,268.53 | 1,688.78 |
| | SVD-LLM | ✓ | 420.25 | 3,716.08 | 1,996.42 |
| | AdaSVD | ✗ | **206.51** | **6,613.44** | **679.66** |
| | AdaSVD | ✓ | **214.51** | **2,728.78** | **654.79** |

### 4.4 INTEGRATE WITH WEIGHT QUANTIZATION

Similar to previous SVD-based compression methods (Hsu et al., 2022a; Yuan et al., 2024; Wang et al., 2025), our AdaSVD is orthogonal to other types of compression techniques. Following Wang et al. (2025), we integrate AdaSVD with the widely used weight quantization method GPTQ (Frantar et al., 2023). As shown in Table 4, we compare AdaSVD with SVD-LLM (Wang et al., 2025) on the LLaMA2-7B model across the WikiText-2, PTB, and C4 datasets. The results demonstrate that, when combined with the 4-bit weight quantization method GPTQ, AdaSVD also consistently outperforms SVD-LLM across all compression ratios. Under high compression ratios (*i.e.*, 60%, 70%, and 80%), AdaSVD + GPTQ-INT4 even surpasses SVD-LLM.

## 5 CONCLUSION

In this work, we propose AdaSVD, an adaptive SVD-based compression method for LLMs. AdaSVD first proposes **adaComp**, which adaptively compensates for the error caused by the truncation of singular matrices, efficiently reducing compression error without requiring additional training. Furthermore, AdaSVD proposes **adaCR**, which adaptively assigns compression ratios based on the importance of each layer, further enhancing performance while maintaining the same target compression rate. Both strategies effectively minimize SVD compression errors, particularly at high compression ratios. Our experiments on multiple open-source LLM and VLM families demonstrate that AdaSVD pushes the performance boundary beyond the current state-of-the-art SVD-based LLM compression methods.

## ETHICS STATEMENT

The research conducted in the paper conforms, in every respect, with the ICLR Code of Ethics.

## REPRODUCIBILITY STATEMENT

We have provided implementation details in Sec. 4. We will also release all the code and models.

## LLM USAGE STATEMENT

Large Language Models (LLMs) were used solely for polishing writing. They did not contribute to the research content or scientific findings of this work.

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
