# AdaSVD: Adaptive Singular Value Decomposition for Large Language Models

## 1 Update Formulas for Matrices U and V

Considering the following SVD compression loss:

$$
\begin{aligned}
\mathcal{L}_{\text{SVD}} &= ||\widehat{W}X - WX||_F^2, \\
&= ||U_k^\sigma (V_k^\sigma)^\top X - WX||_F^2, \\
&= \text{tr}\left((U_k^\sigma (V_k^\sigma)^\top X - WX)^\top (U_k^\sigma (V_k^\sigma)^\top X - WX)\right), \\
&= \text{tr}\left(X^\top V_k^\sigma (U_k^\sigma)^\top U_k^\sigma (V_k^\sigma)^\top X - 2X^\top V_k^\sigma (U_k^\sigma)^\top WX + X^\top W^\top WX\right).
\end{aligned}
\tag{1}
$$

Given the derivative properties:

$$
\frac{\partial \text{tr}(A^\top B)}{\partial A} = B,
\tag{2}
$$

$$
\frac{\partial \text{tr}(ABA^\top)}{\partial A} = A(B + B^\top),
\tag{3}
$$

$$
\frac{\partial \text{tr}(ABA^\top C)}{\partial A} = CAB + C^\top AB^\top,
\tag{4}
$$

$$
\frac{\partial \text{tr}(ABC)}{\partial A} = \frac{\partial \text{tr}(CAB)}{\partial A} = \frac{\partial \text{tr}(BCA)}{\partial A}.
\tag{5}
$$

Taking the partial derivative of $U_k^\sigma$, for the first term:

$$
\frac{\partial}{\partial U_k^\sigma} \text{tr}\left(X^\top V_k^\sigma (U_k^\sigma)^\top U_k^\sigma (V_k^\sigma)^\top X\right) = \frac{\partial}{\partial U_k^\sigma} \text{tr}\left(U_k^\sigma (V_k^\sigma)^\top XX^\top V_k^\sigma (U_k^\sigma)^\top\right),
\tag{6}
$$

let $B = (V_k^\sigma)^\top XX^\top V_k^\sigma$, then we have,

$$
\begin{aligned}
\frac{\partial}{\partial U_k^\sigma} \text{tr}\left(U_k^\sigma B (U_k^\sigma)^\top\right) &= U_k^\sigma (B + B^\top), \\
&= 2U_k^\sigma (V_k^\sigma)^\top XX^\top V_k^\sigma.
\end{aligned}
\tag{7}
$$

For the second term:

$$
\frac{\partial}{\partial U_k^\sigma} \text{tr}\left(-2X^\top V_k^\sigma (U_k^\sigma)^\top WX\right) = \frac{\partial}{\partial U_k^\sigma} \text{tr}\left(-2(U_k^\sigma)^\top WXX^\top V_k^\sigma\right),
\tag{8}
$$

let $B = WXX^\top V_k^\sigma$, we obtain:

$$
\begin{aligned}
\frac{\partial}{\partial U_k^\sigma} \text{tr}\left(-2(U_k^\sigma)^\top B\right) &= -2B, \\
&= -2WXX^\top V_k^\sigma.
\end{aligned}
\tag{9}
$$

Since the third term is independent of $U_k^\sigma$, its gradient is zero.

Thus, the gradient of the SVD loss with respect to $U_k^\sigma$ is:

$$
\begin{aligned}
\frac{\partial \mathcal{L}_{\text{SVD}}}{\partial U_k^\sigma} &= 2U_k^\sigma (V_k^\sigma)^\top XX^\top V_k^\sigma - 2WXX^\top V_k^\sigma, \\
&= U_k^\sigma (V_k^\sigma)^\top XX^\top V_k^\sigma - WXX^\top V_k^\sigma.
\end{aligned}
\tag{10}
$$

Set it to zero, we obtain

$$U_k^\sigma (V_k^\sigma)^\top X X^\top V_k^\sigma = W X X^\top V_k^\sigma,$$
$$\Rightarrow U_k^\sigma = W X X^\top V_k^\sigma ((V_k^\sigma)^\top X X^\top V_k^\sigma)^{-1}. \tag{11}$$

Similarly, taking the partial derivative of $(V_k^\sigma)^\top$, for the first term:

$$\frac{\partial}{\partial V_k^{\sigma\top}} \mathrm{tr}\Big( X^\top V_k^\sigma (U_k^\sigma)^\top U_k^\sigma (V_k^\sigma)^\top X \Big) = \frac{\partial}{\partial V_k^{\sigma\top}} \mathrm{tr}\Big( (V_k^\sigma)^\top X X^\top V_k^\sigma (U_k^\sigma)^\top U_k^\sigma \Big), \tag{12}$$

let $B = X X^\top$ and $C = (U_k^\sigma)^\top U_k^\sigma$, then we have

$$\frac{\partial}{\partial V_k^{\sigma\top}} \mathrm{tr}\Big( (V_k^\sigma)^\top B V_k^\sigma C \Big) = C V_k^{\sigma\top} B + C^\top V_k^{\sigma\top} B^\top,$$
$$= 2 (U_k^\sigma)^\top U_k^\sigma V_k^{\sigma\top} X X^\top. \tag{13}$$

For the second term:

$$\frac{\partial}{\partial V_k^{\sigma\top}} \mathrm{tr}\Big( -2 X^\top V_k^\sigma (U_k^\sigma)^\top W X \Big) = \frac{\partial}{\partial V_k^{\sigma\top}} \mathrm{tr}\Big( -2 V_k^\sigma (U_k^\sigma)^\top W X X^\top \Big), \tag{14}$$

let $B = (U_k^\sigma)^\top W X X^\top$, we obtain:

$$\frac{\partial}{\partial V_k^{\sigma\top}} \mathrm{tr}\Big( -2 V_k^\sigma B \Big) = -2B,$$
$$= -2 (U_k^\sigma)^\top W X X^\top. \tag{15}$$

Since the third term is independent of $V_k^{\sigma\top}$, its gradient is zero.

Thus, the gradient of the SVD loss with respect to $V_k^{\sigma\top}$ is:

$$\frac{\partial \mathcal{L}_{\mathrm{SVD}}}{\partial V_k^{\sigma\top}} = 2 (U_k^\sigma)^\top U_k^\sigma V_k^{\sigma\top} X X^\top - 2 (U_k^\sigma)^\top W X X^\top,$$
$$= (U_k^\sigma)^\top U_k^\sigma V_k^{\sigma\top} X X^\top - (U_k^\sigma)^\top W X X^\top. \tag{16}$$

Set it to zero, we obtain

$$(U_k^\sigma)^\top U_k^\sigma V_k^{\sigma\top} X X^\top = (U_k^\sigma)^\top W X X^\top,$$
$$\Rightarrow V_k^{\sigma\top} = ((U_k^\sigma)^\top U_k^\sigma)^{-1} (U_k^\sigma)^\top W X X^\top (X X^\top)^{-1} = ((U_k^\sigma)^\top U_k^\sigma)^{-1} (U_k^\sigma)^\top W. \tag{17}$$

## 2 PSEUDOCODE OF AdaSVD

Algorithm 1 shows the pseudocode of AdaSVD. AdaSVD first applies stack-of-batch strategy to the calibration data $C$, then runs the adaptive compression ratio calculation as shown in Algorithm 2 to obtain layerwise compression ratio. After that, AdaSVD runs the SVD decomposition and truncation with the importance-aware compression ratio on each weight matrix in the LLM. To further compensate for the compression error, AdaSVD also runs the adaptive compensation for matrix $\mathcal{U}$ and $\mathcal{V}^\top$, as shown in Algorithm 3.

## 3 CONTENTS GENERATED FROM THE MODEL COMPRESSED BY AdaSVD AND SVD-LLM

We compare some examples of sentences generated by LLaMA-2-7B compressed with AdaSVD and SVD-LLM (Wang et al., 2025) in Table 1. As demonstrated, the sentences produced by the model compressed using AdaSVD show enhanced smoothness, significance, and informativeness compared to those compressed by SVD-LLM. Moreover, when the compression ratio reaches 40%, the previously leading method, SVD-LLM, begins to fail in generating coherent sentences. However, even when the compression ratio is up to 50%, AdaSVD is still capable of generating logical and meaningful sentences.

---

**Algorithm 1** Pseudocode of `AdaSVD`

---

1: **Inputs:** LLM $\mathcal{M}$, Calib Data $\mathcal{C}$, Bucket Size $M$, Target Retention Ratio $trr$, Min Retention Ratio $mrr$, Update Iteration $k$
2: **Outputs:** Updated Model $\mathcal{M}'$ by `AdaSVD`
3: **procedure** ADASVD($\mathcal{M}, \mathcal{C}, trr, mrr, k$)
4:      $\mathcal{X} \leftarrow$ GET_CALIB($\mathcal{C}$)             $\triangleright$ Randomly collect samples as calibration data
5:      $\mathcal{X}'[1], \mathcal{X}'[2], ..., \mathcal{X}'[M] \leftarrow$ SOB($\mathcal{X}, M$)      $\triangleright$ Shuffle samples and utilize SOB strategy
6:      Set$_{\mathcal{S}} \leftarrow$ WHITENING($\mathcal{M}, \mathcal{X}'$), Set$_{\mathcal{SVD}} \leftarrow \emptyset$, Set$_{\mathcal{W}} \leftarrow \mathcal{M}$
7:      Set$_{\mathcal{CR}} \leftarrow$ LAYER_CR($\mathcal{M}, \mathcal{X}', trr, mrr$)          $\triangleright$ Measure layerwise importance
8:      **for** layer $i$ **in** language model $\mathcal{M}$ **do**
9:          $\mathcal{W}_i \leftarrow$ Set$_{\mathcal{W}}(i)$, $\mathcal{S}_i \leftarrow$ Set$_{\mathcal{S}}(\mathcal{W}_i)$    $\triangleright$ Extract the whitening matrix of current weight $\mathcal{W}_i$
10:        $\mathcal{U}_i, \Sigma_i, \mathcal{V}_i \leftarrow$ SVD($\mathcal{W}_i \mathcal{S}_i$)          $\triangleright$ Apply Singular Value Decomposition
11:        $\Sigma' \leftarrow$ TRUNC($\Sigma_i$), $(\mathcal{U}_i', \mathcal{V}_i') \leftarrow$ TRUNC_UV($\mathcal{U}, \mathcal{V}, \Sigma'$)      $\triangleright$ Truncate with adaptive ratio
12:        Set$_{\mathcal{SVD}} \leftarrow (\mathcal{U}_i', \mathcal{V}_i') \cup$ Set$_{\mathcal{SVD}}$
13:      **end for**
14:      $\mathcal{M}' \leftarrow$ ADA_UPDATE($\mathcal{M}, \mathcal{X}'$, SET$_{\mathcal{SVD}}, k$)      $\triangleright$ Alternate update $\mathcal{U}_i', \mathcal{V}_i'$ for $k$ iterations
15:      **return** $\mathcal{M}'$
16: **end procedure**

---

**Algorithm 2** Pseudocode of Layerwise Adaptive Compression Ratio

---

1: **Inputs:** LLM $\mathcal{M}$, SOB Preprocessed Calib Data $\mathcal{X}$, Target Retention Ratio $trr$, Min Retention Ratio $mrr$
2: **Outputs:** Set$_{\mathcal{CR}}$: Set of compression ratios for each layer in $\mathcal{M}$
3: **procedure** LAYER_CR($\mathcal{M}, \mathcal{X}, trr, mrr$)
4:      Set$_{\mathcal{CR}} \leftarrow \emptyset$             $\triangleright$ Initialize the set of compression ratio
5:      $\mathcal{I}(\mathcal{W}) \leftarrow \emptyset$           $\triangleright$ Initialize the set of layerwise importance
6:      Set$_{\mathcal{W}} \leftarrow \mathcal{M}$             $\triangleright$ Obtain weights of each layer in $\mathcal{M}$
7:      **for** $i = 1, 2, \ldots,$#layers **do**
8:        $\mathcal{W} \leftarrow$ Set$_{\mathcal{W}}[i]$, $\mathcal{Y} \leftarrow \mathcal{W}\mathcal{X}$
9:        $\mathcal{I}(\mathcal{W}) \leftarrow \frac{\mathcal{W}\mathcal{X}}{|\mathcal{W}||\mathcal{X}|} \cap \mathcal{I}(\mathcal{W})$ $\triangleright$ Utilize cosine similarity to calculate the importance of each layer
10:        $\mathcal{X} \leftarrow \mathcal{Y}$
11:      **end for**
12:      $\mathcal{I}_n(\mathcal{W}) \leftarrow \frac{\mathcal{I}(\mathcal{W})}{\text{mean}\mathcal{I}(\mathcal{W})}$          $\triangleright$ Normalize $\mathcal{I}(\mathcal{W})$ by mean value
13:      Set$_{\mathcal{CR}} \leftarrow mrr + \mathcal{I}_n(\mathcal{W}) \cdot (trr - mrr)$ $\triangleright$ Compression ratios of each layer based on relative importance
14:      **return** Set$_{\mathcal{CR}}$
15: **end procedure**

---

## 4 MORE VISUALIZATION OF ADAPTIVE COMPENSATION

Figures 1 and 2 shows more visualization results of output distribution before and after adaptive compensation, which includes gate_projection, up_proj, v_proj and o_proj layers in Llama7B, Llama2-7B and Vicuna-7B etc. We can observe that after adaptive Compensation, the distribution of outputs quickly converges with the original output, and the overlap area of two distributions is significantly improved, demonstrating a good compensation effect.

## 5 MORE VISUALIZATION OF LAYER-WISE RELATIVE IMPORTANCE

Figure 3 shows that the importance of different layers varies. It can be observed that the first layer always weighs the most importance, suggesting that we should retain more weight on it. For the Llama, Mistral and Vicuna family, the relative importance curve approximates a bowl shape, highlighting the significance of both the initial and final layers. While for Opt family, the first layer normally holds greatest importance.

**Algorithm 3** Pseudocode of Layerwise Adaptive Compensation Update

1: **Inputs:** LLM $\mathcal{M}$, SOB Preprocessed Calib Data $\mathcal{X}'$, Whitening Matrices $\text{Set}_S$, Decomposed Matrices for Weights $\text{Set}_{\mathcal{SVD}}$
2: **Outputs:** $\mathcal{M}'$: Compressed LLM by `AdaSVD`
3: **procedure** ADA_UPDATE($\mathcal{M}, \mathcal{X}', \text{Set}_S, \text{Set}_{\mathcal{SVD}}$)
4:    $\mathcal{M}' \leftarrow \mathcal{M}$                                                                        ▷ Initialize $\mathcal{M}'$ with $\mathcal{M}$
5:    $\text{Set}_{\mathcal{L}} \leftarrow \mathcal{M}'$                           ▷ Obtain the set of encoder and decoder layers in $\mathcal{M}'$
6:    $\mathcal{X}' \leftarrow \mathcal{M}'(\mathcal{X}')$                              ▷ Obtain the input activation of the first layer in $\mathcal{M}'$
7:    **for** $\mathcal{L}$ **in** $\text{Set}_{\mathcal{L}}$ **do**
8:        **for** iter $= 1, 2, \ldots, \tau$ **do**
9:            $\text{Set}_{\mathcal{W}} \leftarrow \mathcal{L}$                         ▷ Obtain the set of weights in $\mathcal{L}$ to compress
10:           **for** $\mathcal{W}$ **in** $\text{Set}_{\mathcal{W}}$ **do**
11:               $\mathcal{S} \leftarrow \text{Set}_S(\mathcal{W})$                                  ▷ Obtain whitening results
12:               $\mathcal{U}, \Sigma, \mathcal{V} \leftarrow \text{Set}_{\mathcal{SVD}}(\mathcal{W})$       ▷ Obtain the decomposed matrices of $\mathcal{W}$
13:               $\mathcal{U}' \leftarrow (\mathcal{V}_A \Sigma_A^+ \mathcal{U}_A^\top \mathcal{B})^\top$             ▷ Adaptive compensation for U
14:               $\mathcal{V}'^\top \leftarrow \left( (\mathcal{U})^+ \right)^\top \mathcal{W}$                  ▷ Adaptive compensation for V
15:               $\mathcal{L}(\mathcal{W}) \leftarrow \mathcal{L}(\mathcal{U}', \mathcal{V}'^\top)$       ▷ Replace $W$ with updated $\mathcal{U}'$ and $\mathcal{V}'^\top$ in $\mathcal{L}$
16:           **end for**
17:           $\mathcal{X}' \leftarrow \mathcal{L}(\mathcal{X}')$       ▷ Use the compressed layer to calculate the new input activation $\mathcal{X}'$
18:       **end for**
19:   **end for**
20:   **return** $\mathcal{M}'$
21: **end procedure**

Table 1: An example of contents generated by the compressed LLaMA-2-7B-hf by `AdaSVD` and SVD-LLM (Wang et al., 2025) under different compression ratios. The input is marked in bold and the normal texts are the generated sentences.

| RATIO | ADASVD | SVD-LLM (WANG ET AL., 2025) |
|---|---|---|
| Original | **What is a computer?** A computer is an electronic device that can be programmed to carry out sequences of arithmetic or logical operations automatically. The modern computer can perform all the functions of a calculator, but it can also do much more. It can store and retrieve information, process data, and communicate with other computers... | |
| 20% | **What is a computer?** A computer is an electronic device that can be programmed to perform tasks. It has the ability to store information, process information and communicate with other computers. Computers are used in many industries... | **What is a computer?** A computer is a device that can be used to store information. A computer can also be connected to other devices, such as a printer or an inkjet printer. In this way, it is possible to use the printers to print images on paper... |
| 30% | **What is a computer?** A computer is an electronic device that can be used to perform calculations. Computers have been used for scientific research, astronomical observations, meteorological studies, geophysical investigations and other scientific activities... | **What is a computer?** The computer is a computer.The 1960s, the 20th century.The computer was invented in the late 50's and early 70 's. It was not used until the end of the nineteenth century when it became widely used. Computers were introduced to the public... |
| 40% | **What is a computer?** A computer is an electronic device that can be used as a calculator. It was invented in the 1950s and has been widely used since then. Computers are used to perform calculations, mathematical calculations. They are also used for scientific research... | **What is a computer?** The machine, it is a computer.The computer is an electronic device. The machine is mechanical. A mechanical device is made of mechanical devices. Mechanical devices are mechanical machines... |
| 50% | **What is a computer?** The first computer was called a Transitor. In the late 70's and early 80's, the Transitors were replaced by computers known as Personal Computers. At this time, these machines could be used for scientific research and ... | **What is a computer?** What is a computer? t, t , t . The t of t he t h t e t t a t at t ta tatatataaataataasaasaaseaseaeaeaaseaseaesease... |

## 6 MORE RESULTS OF HIGHER COMPRESSION RATIOS AND ABLATION STUDY

Table 2 compares `AdaSVD` with previous SVD-based compression methods under 70% and 80% compression ratios, where `AdaSVD` exhibits much more performance gain with higher compression demand. Table 3 shows the comprehensive results of our ablation study, especially the results from

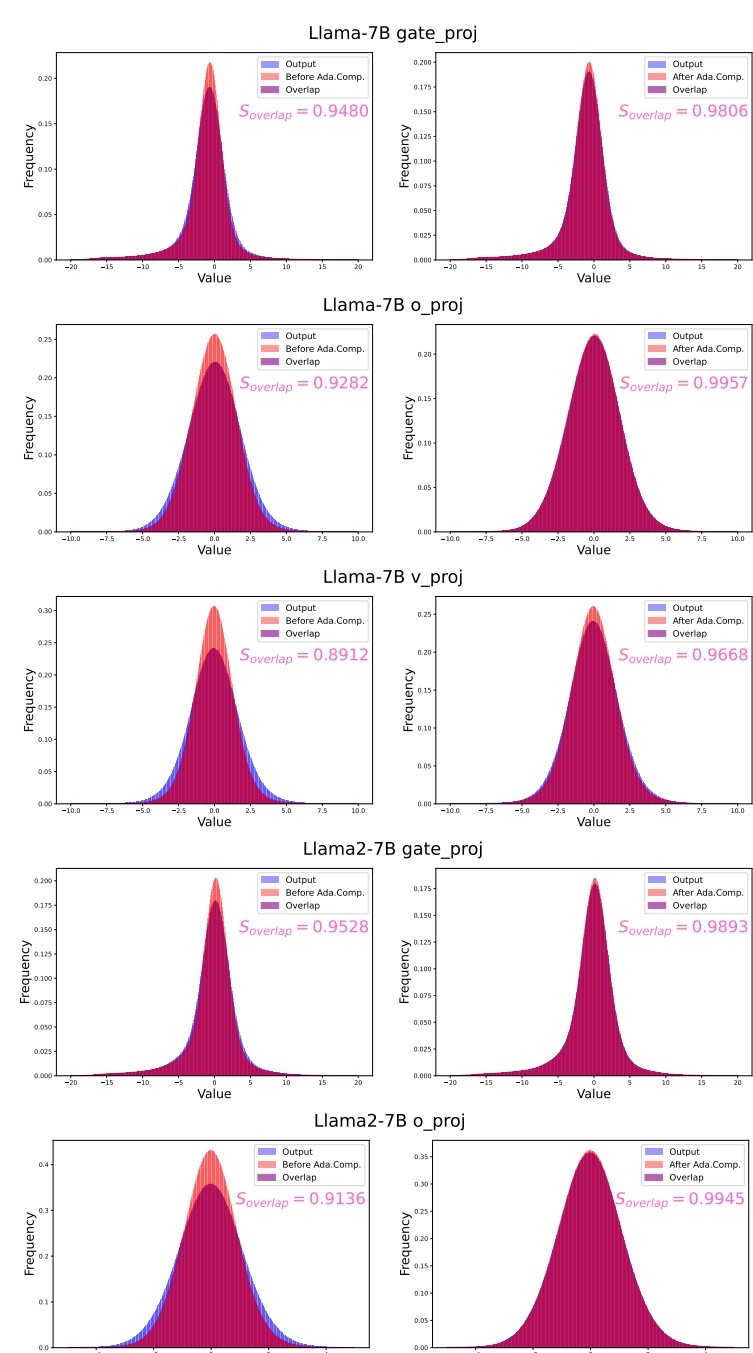

Figure 1: More Visualization of Adaptive Compensation.(i) **Left:** Output distribution before adaptive compensation. **Right:** Output distribution after adaptive compensation.

SVD-LLM and `AdaSVD` under 70% and 80% compression ratio, where our method also demonstrates a large advantage.

# 7 MORE RESULTS OF IMAGE CAPTION BY LLAVA MODEL ON COCO DATASET

Figure 4, Figure 5 and Figure 6 show more results of image caption from language models in LLaVA-7B under 40%, 50% and 60% compression ratio respectively and they also compare the performance

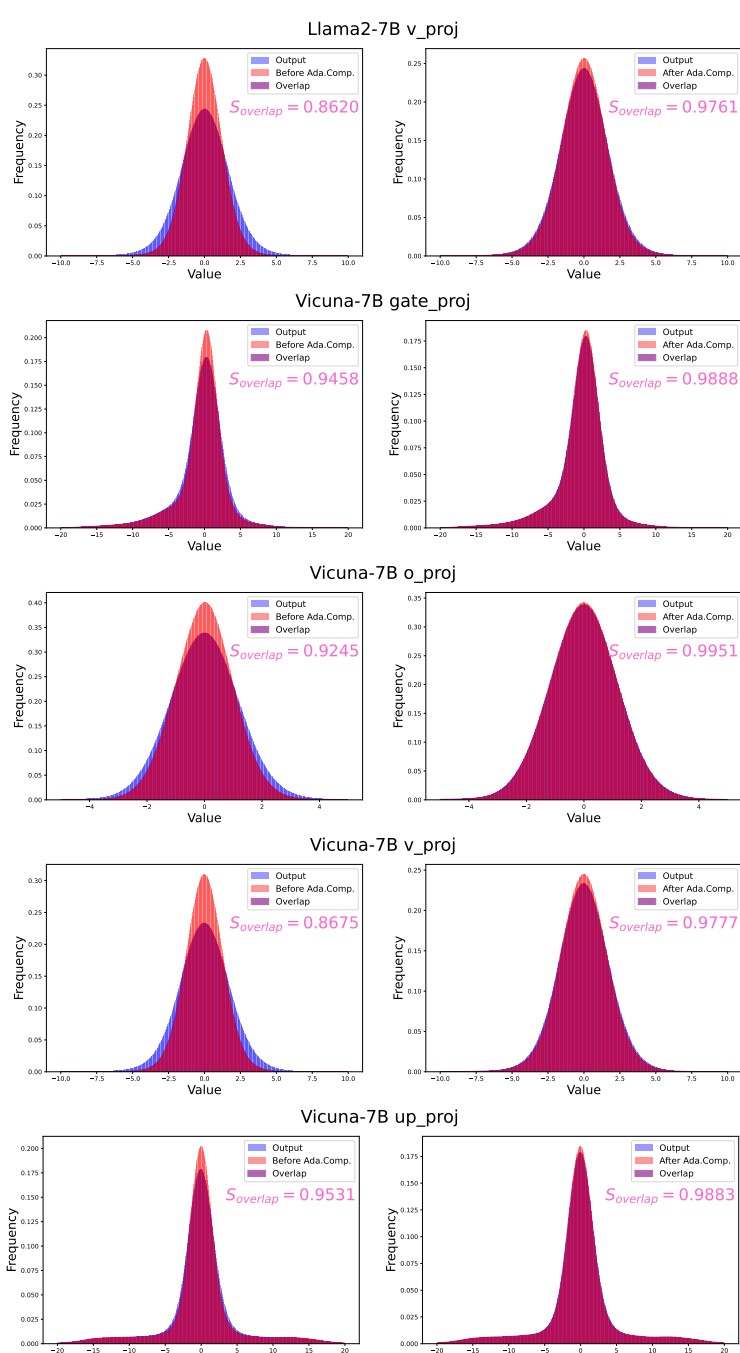

Figure 2: More Visualization of Adaptive Compensation.(ii) **Left:** Output distribution before adaptive compensation. **Right:** Output distribution after adaptive compensation.

differences among SVD, SVDLLM and AdaSVD on COCO dataset. We use question "What is in the picture?" as an example and highlight the correct captions and wrong captions in different colors.

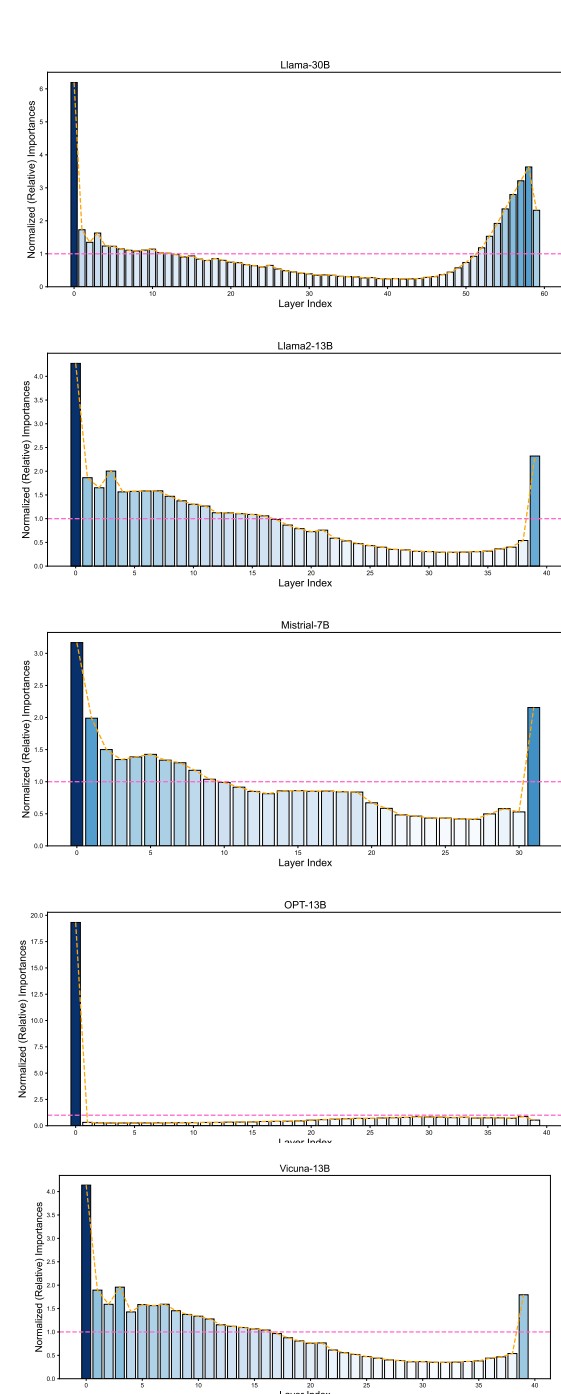

Figure 3: More Visualization of Layer-wise Relative Importance.

Table 2: Perplexity of LLaMA2-7B between `AdaSVD` and previous SVD compressed methods under 70% and 80% compression ratios.

| Ratio | Method | WikiText-2↓ | PTB↓ | C4↓ |
|-------|--------|-------------|------|-----|
| 0% | Original | 5.68 | 8.35 | 7.34 |
| 70% | SVD | 40,010.99 | 333,773.00 | 55,284.00 |
| | FWSVD (Hsu et al., 2022) | 28,769.57 | 27,470.32 | 31,147.58 |
| | ASVD (Yuan et al., 2024) | 28,916.46 | 19,463.15 | 17,615.46 |
| | SVD-LLM (Wang et al., 2025) | 125.16 | 6,139.78 | 677.38 |
| | **AdaSVD** | **96.38** (↓ 23%) | **1596.55** (↓ 74%) | **411.22** (↓ 39%) |
| 80% | SVD | 24,964.78 | 41,571.00 | 53,894.00 |
| | FWSVD (Hsu et al., 2022) | 9,482.33 | 7,133.20 | 11,058.99 |
| | ASVD (Yuan et al., 2024) | 15,000.25 | 12,134.83 | 16,716.60 7 |
| | SVD-LLM (Wang et al., 2025) | 372.48 | 6,268.53 | 1,688.78 |
| | **AdaSVD** | **200.43** (↓ 46%) | **4246.36** (↓ 32%) | **730.74** (↓ 57%) |

Table 3: Ablation study on LLaMA-2-7B. Results are measured by perplexity, with best results highlighted in ▢ .

(a) Effectiveness of Adaptive Compensation

| Method | Tgt. CR | adaComp | WikiText2↓ | C4↓ |
|--------|---------|---------|------------|-----|
| SVD-LLM | 40% | ✗ | 16.11 | 61.95 |
| AdaSVD | 40% | ✗ | 15.42 | 65.50 |
| AdaSVD | 40% | ✓ | 14.76 | 56.98 |
| SVD-LLM | 50% | ✗ | 27.19 | 129.66 |
| AdaSVD | 50% | ✗ | 27.33 | 126.85 |
| AdaSVD | 50% | ✓ | 25.58 | 113.84 |
| SVD-LLM | 60% | ✗ | 89.90 | 561.00 |
| AdaSVD | 60% | ✗ | 78.82 | 339.31 |
| AdaSVD | 60% | ✓ | 60.08 | 294.26 |
| SVD-LLM | 70% | ✗ | 125.16 | 677.38 |
| AdaSVD | 70% | ✗ | 140.30 | 589.75 |
| AdaSVD | 70% | ✓ | 107.90 | 441.33 |
| SVD-LLM | 80% | ✗ | 372.48 | 1688.78 |
| AdaSVD | 80% | ✗ | 301.16 | 1123.27 |
| AdaSVD | 80% | ✓ | 206.51 | 679.66 |

(b) Effectiveness of Adaptive Compression Ratio

| Method | Tgt. CR | CR | WikiText2↓ | C4↓ |
|--------|---------|----|------------|-----|
| SVD-LLM | 40% | Const | 16.11 | 61.95 |
| AdaSVD | 40% | Const | 15.42 | 65.50 |
| AdaSVD | 40% | Adapt | 14.85 | 57.08 |
| SVD-LLM | 50% | Const | 27.19 | 129.66 |
| AdaSVD | 50% | Const | 27.33 | 126.85 |
| AdaSVD | 50% | Adapt | 26.01 | 117.58 |
| SVD-LLM | 60% | Const | 89.90 | 561.00 |
| AdaSVD | 60% | Const | 69.46 | 336.90 |
| AdaSVD | 60% | Adapt | 60.08 | 294.26 |

(c) Iteration Number for Adaptive Compression

| Method | Tgt. CR | #Iter | WikiText2↓ | C4↓ |
|--------|---------|-------|------------|-----|
| SVD-LLM | 40% | - | 16.11 | 61.95 |
| AdaSVD | 40% | 1 | 14.85 | 57.08 |
| AdaSVD | 40% | 3 | 15.47 | 57.28 |
| AdaSVD | 40% | 15 | 15.84 | 57.39 |
| SVD-LLM | 50% | - | 27.19 | 129.66 |
| AdaSVD | 50% | 1 | 26.01 | 117.58 |
| AdaSVD | 50% | 3 | 27.11 | 115.51 |
| AdaSVD | 50% | 15 | 27.45 | 110.35 |
| SVD-LLM | 60% | - | 89.90 | 561.00 |
| AdaSVD | 60% | 1 | 60.08 | 294.26 |
| AdaSVD | 60% | 3 | 64.12 | 301.19 |
| AdaSVD | 60% | 15 | 62.34 | 267.29 |
| SVD-LLM | 70% | - | 125.16 | 677.38 |
| AdaSVD | 70% | 1 | 140.30 | 589.75 |
| AdaSVD | 70% | 3 | 121.66 | 474.01 |
| AdaSVD | 70% | 15 | 107.90 | 441.33 |
| SVD-LLM | 80% | - | 372.48 | 1688.78 |
| AdaSVD | 80% | 1 | 301.15 | 1123.27 |
| AdaSVD | 80% | 3 | 233.20 | 793.30 |
| AdaSVD | 80% | 15 | 206.51 | 679.66 |

(d) Minimum Retention Ratio for Adaptive CR

| Method | Tgt. CR | MRR | WikiText2↓ | C4↓ |
|--------|---------|-----|------------|-----|
| SVD-LLM | 40% | - | 16.11 | 61.95 |
| AdaSVD | 40% | 0.40 | 15.01 | 57.17 |
| AdaSVD | 40% | 0.45 | 14.85 | 57.08 |
| AdaSVD | 40% | 0.50 | 14.76 | 56.98 |
| SVD-LLM | 50% | - | 27.19 | 129.66 |
| AdaSVD | 50% | 0.40 | 25.58 | 113.84 |
| AdaSVD | 50% | 0.45 | 26.01 | 117.58 |
| AdaSVD | 50% | 0.50 | 27.33 | 126.85 |
| SVD-LLM | 60% | - | 89.90 | 561.00 |
| AdaSVD | 60% | 0.30 | 50.33 | 239.18 |
| AdaSVD | 60% | 0.35 | 53.17 | 256.66 |
| AdaSVD | 60% | 0.40 | 60.08 | 294.26 |
| SVD-LLM | 70% | - | 125.16 | 677.38 |
| AdaSVD | 70% | 0.20 | 96.38 | 411.22 |
| AdaSVD | 70% | 0.25 | 103.50 | 438.09 |
| AdaSVD | 70% | 0.30 | 107.90 | 441.33 |
| SVD-LLM | 80% | - | 372.48 | 1688.78 |
| AdaSVD | 80% | 0.10 | 204.72 | 759.86 |
| AdaSVD | 80% | 0.15 | 200.43 | 730.74 |
| AdaSVD | 80% | 0.20 | 206.51 | 679.66 |

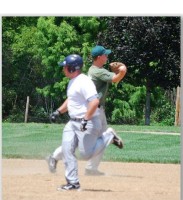

**SVD**:  says that he's a man who has no knowledge of what he wants to do, but he does not know it and his father was killed by an air raid. The other person on the field is wearing green clothes ...

**SVD-LLM**: A man standing on his knee, he has a knife - an object that can be used to cut down trees and forests. The killer' ' s head was not seen by the people who were present at the time of the event, but it was shown in front of many people, one of them was a person who was wearing a hat ...

**AdaSVD**: The image of a man playing baseball on a field surrounded by trees and bushes. He's wearing his uniform, we wears an outfit that he we can wear to play baseball, ...

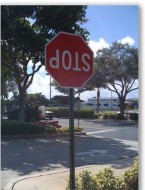

**SVD**:  The city'in Florida has a large number of residents who have been affected by hurricanes and other natural disasters. In Florida, many residents were displaced from their homes by Hurricane

**SVD-LLM**: The sign on the road says that it's a large truck, which has been used to move heavy loads of vehicles. The sign also states that this vehicle is an old one and its wheels are not very big but they are larger than the other vehicles we have seen. It'tts much more powerful than any other vehicle ...

**AdaSVD**: It's a street sign that has letters on it. The letters are written on the sides of the road ...

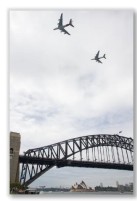

**SVD**:  - As an example, it's a large and complex book that has been used by many of the most well - known authors, including T. - S - Eli - El - Ell - who was one of America' - s - most influential poets...

**SVD-LLM**: - As a result, it's an all - to - be - as - well - and - has - its - highest - peak - at - 1200 - feet - above - the - top - of - mountains - where - there - are - many - mountain - peaks - that - have ...

**AdaSVD**: The airplane flies over the city, flying at a height of 100 feet above the bridge. It's an aeroplane that has been used by the military during World War II and was used as a transport aircraft for the United States Air Force during the Vietnam War. ...

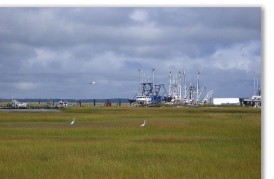

**SVD**:  I'm a person who has been shown to be one of the most well - known and respected people in this country\n\nI' 's work was done on his own land, which was used by him for many years, he had  ...

**SVD-LLM**: 2013 - I have a lot of other things, it's an old man who has been there for many years, and he says that we are not going to be able to do anything at all. - He is very uncomforted by his wife, but she does not want him to go away from his family, which is one of the most important people in ...

**AdaSVD**: There's a lot of birds flying over the area, and there are many birds on the beach. The sky is covered by clouds that have been blown away by the winds from the sea, ...

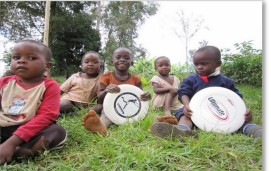

**SVD**:  it's not a game that can be played by children and adults. It' 's an unplayed game of soccer, which has been played for many generations, but it is not played on the field, this game is played  ...

**SVD-LLM**: a group of children are seen playing with an object that is placed on the ground, it is shown to be used as a symbol of the people'ans who have not been present in this area. The other person has no knowledge of his presence, but he does not know what he knows ...

**AdaSVD**:  There are a group of children playing on the ground, they are wearing red cloths and white cloth. The children are dressed in their clothes, ...clothing is different from the color of their skin, ...

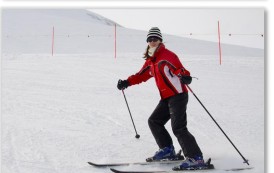

**SVD**: She is wearing a red coat, and she wears a hat on her head. She's dressed in snowshoe boots, which are made of snow shoes that are worn by snowboats, they are used for skiers ...

**SVD-LLM**: a woman wears a red hat, she's wearing her hat on her head, and she is weared with a black coat, which she we was weareed by an old man who has worn his hat for many decades ...

**AdaSVD**:  The person is a woman, dressed in her red coat and wearing a hat on her head. She is walking through the snow with her skate skiers. ...

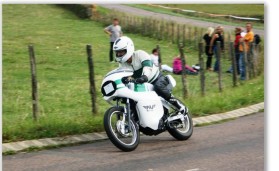

**SVD**: The man wears a helmet, and he's wearing a hat on his head. ...

**SVD-LLM**: a man on a motorcycle, who is wearing a white suit and a black hat, he is seen to be walking with a group of people, some of whom are dressed in pink clothes ...

**AdaSVD**:  The man rides a motorcycle, wearing a helmet on his head and we wears an out - of - clothes , which he we wore during the race at the end of the season. In this scene, the man's motorcycle is shown driving through the road with people who are watching him ... He is dressed in a suit and has a hat on top of his hat ...

Figure 4: We perform more image captioning by applying SVD, SVD-LLM, and our `AdaSVD` to LLaVA-7B model on the COCO dataset respectively under 40% compression ratio, highlighting the correct captions and wrong captions in different colors.

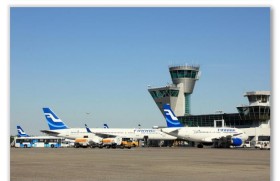

**SVD**: of London and its surrendings are situated on the river. In this city, it is surrounded by a number of rivers that run through the Thames and their tributaries are located along the River'ans nearby ...

**SVD-LLM**: of a man who has been seen by many people, it is not possible to be able to do anything that he can do. It's does not have done any work on him, and no one knows what he will do, ...

**AdaSVD**: The city of Westminster is surrounded by the river and it's a large part of the city. It is situated on the Thame River, which is an important river that runs through London and has been used as a source of water for many years. In this city there are two bridges over the rivers, one bridge is located at the Tame Bridge and another bridge at Chester Bridge ...

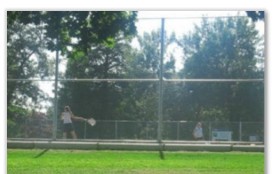

**SVD**: was used by the British Army during the Second World War. In 1942, the RAF had a total of 300 aircraft and 50 pil ...

**SVD-LLM**: - a man who was an officer of the British Army, he had been captured by the Germans at the beginning of his career. He was one of those officers who were killed during World War II, and it was also known to have been involved in various battles against the Italians The Italans were ...

**AdaSVD**: The motor vehicle was used by the British Air Force during the Second World War, and it was also used as a transport vehicle for the RAF ...

**SVD**: has been used by a number of air transport companies, including Air Canada'ans and Canadian Airlines. The aeroplane company operates its own aircraft, as well as other carriers that have their own carriations, such as Air Canadian Airs and Air Transport Airlinas. ...

**SVD-LLM**: - a large number of people are seen to be on this day, and it' has been observed by many people that they have not done their work.\n1980 ...

**AdaSVD**: The airport has a large number of planes, it's an aeroplane that was used by the Air Force during the Second World War. ...

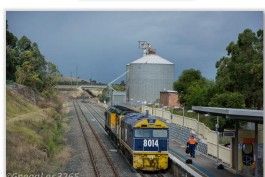

**SVD**: and she is playing her rakas on a tennis court. She plays her game with her opponent, where she has been beaten by an oppositor of her own opponents who are not allowed to play their game. ...

**SVD-LLM**: of a woman who plays tennis on the court, she has been playing it for many decades. She is an old woman that played her first time at the age of 20 years, and she was not able to play tennis with her right arm, but she can play her left arm by hitting her arm against her opponent's arm ...

**AdaSVD**: a woman playing tennis on her tennis court, and she plays tennis with tennis players ...

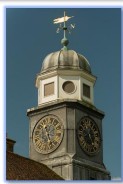

**SVD**: and has a clock on top of the tower. The clocks are located on the roof of St. Peter's church, which was built in 1634, it is one of London 's most well - known churches. ...

**SVD-LLM**: - of a clock, which has been built on the roof of an old building and it is surrounded by two large clocks that are placed at the top of the tower. The clock is situated atop of this tower, with one clock mounted at its base, and another clock located at both sides of it ...

**AdaSVD**: The clocks are on the tower and it's an old clock. The church was built in 1640 ... is one of the most well-known churches in England and has been used as a place of worship for over 300 years ...

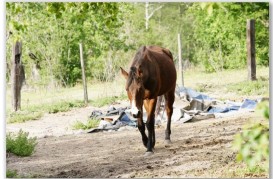

**SVD**: are a train that runs on rail tracks and has been used by the Australian Railways to transport trains. The railway' Railway Railway was built in Victoria in 1892, with its first railway railway ...

**SVD-LLM**: of a man standing on his knees, we has been seen to be one of the most well - known people who have not done much work.\n1980 - 2003 \u2014 1645 ) and he' was an important person in Australia ' s political party that was very popular at the time. He was also involved in ...

**AdaSVD**: The train runs through the railway station, where it' 's a passenger of the rail line. ... It has been used by the people who have made their way from the city ...

**SVD**: of a horse that runs through an open area, it' 's running on the ground. The horse is walking over the land and moving its horses to move their horses are not allowed to be used by people who ...

**SVD-LLM**: a horse that has been cut by an old man, he was not seen to be on his head and feet, it' ' s legs are covered by the bones of the horses who have been used for many decades. The horse is a very large horse with a big number of horses - one horse can be more than 100 miles away from the city ...

**AdaSVD**: The horse is walking through the woods of a farmer' 's house. ... on the ground , which is covered with trees and bushes ...

Figure 5: We perform more image captioning by applying SVD, SVD-LLM, and our `AdaSVD` to LLaVA-7B model on the COCO dataset respectively under 50% compression ratio, highlighting the correct captions and wrong captions in different colors.

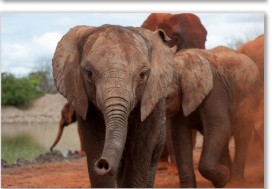

**SVD**: it's a man who is wearing his tennis racket on his arm, and he has been injured by an injury that he was not able to play at the age of 20 years. ...

**SVD-LLM**: , and it' was a man who has been seen to be uncovered on the ground of his foot) He also made an act that he had not done by him.\n1980 - 125 )\n36 \u2013 47 - It was one of the most well known people with this person, ...

**AdaSVD**: It was a man who played tennis ....

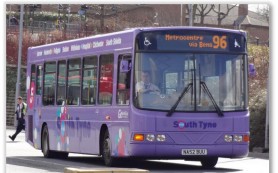

**SVD**: and it's a person who has been shown as an example of what he was seen to be one of the most well - known figures of his time. In this book, there are many characters that have their own ...

**SVD-LLM**: and he has been seen by many people, who have a large number of animals that are not known to be found.) He is one of the most well known animals, it is an animal that is more common than the other animals. ...

**AdaSVD**: it was very large animals that are living on the country ...

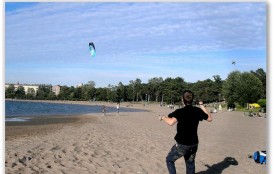

**SVD**: The driver of a truck and his passengers are seen to be walking on the road. In this book, it's described that he has been captured by an old man who was wearing a coat and a hat ...

**SVD-LLM**: and he has been seen to be a person of this city. The vehicle, which is not used by its people, it' was stopped on the way of the vehicles that are an old man who can have no go his life...

**AdaSVD**: it'is a vehicle that has been used by the people of the city. The vehicles are made to be transported on the streets and roads ...

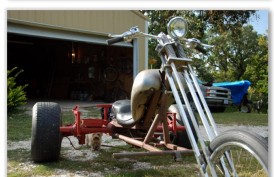

**SVD**: The man's face is seen wearing a black coat, and his hands are shown to be covered by the water. In this scene, the man is shown standing on the beach with his feet covered in sand ...

**SVD-LLM**: ) and it has been used by a man, who was not seen to be found on this day.\n\n ),\n1980 - 200 ; 150 , 346 ) = 70 / 60 ) . The species are known to have been observed as an uncommon person of the species that can be described with its presence, which may be more than one- third or two ...

**AdaSVD**: he was a person who were on the beach, ...

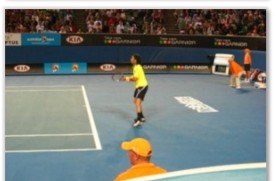

**SVD**: a man standing on one side of the road and another man sitting on the other side. In this book, it'ans was written by an author who wrote that he did not have been able to write his book because ...

**SVD-LLM**: , and it has been used by many people.\n\n Many of them have a large number of animals that are not to be seen on this land. It is also known as being an uncommon man who is very more than one hundred years, but he was found to make his own life. He had no great numbers of men ...

**AdaSVD**: It was a car that was used by the people of the country. ...

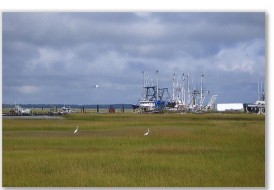

**SVD**: was a man who had his head cut off and he could not be seen, but this man's head was cut to his neck. the people were killed by their heads and they were thrown out of their sk ...

**SVD-LLM**: , who has been seen by a man of his own person. was not found to be one of the most people that he had an uncovered on this day, and it was also known to have no more than two - ...

**AdaSVD**: was a person of an Australian tennis player who played his tennis at the stadium. ...

**SVD**: I'm a person who has been shown to be one of the most well - known and respected people in this country I' 's work was done on his own land, which was used by him for many years, he had ...

**SVD-LLM**: ) and it has been found to be an uncovered by a large number of people.\n )\n The birds are not known to have been seen, that this bird' was observed by many animals \" - I ' s\" - \" ; \" .\n\n ...

**AdaSVD**: ... birds were flying on the beach. The birds of birds flew to fly their wings , ... it was found that the birds would have flowed over the sea for two days ...

Figure 6: We perform more image captioning by applying SVD, SVD-LLM, and our `AdaSVD` to LLaVA-7B model on the COCO dataset respectively under 60% compression ratio, highlighting the correct captions and wrong captions in different colors.