# OpenReview forum: "AdaSVD: Adaptive Singular Value Decomposition for Large Language Models"
_ICLR.cc/2026/Conference — ICLR 2026 Conference Withdrawn Submission_

### Official Review · Reviewer_X4eT · 2025-10-27

**Soundness:** 2
**Presentation:** 3
**Contribution:** 1
**Rating:** 2
**Confidence:** 5

**Summary:**

To mitigate the model performance degradation after model decomposition (i.e., Singular Values Decomposition), authors propose a adaptive SVD-based LLM compression approach. Specifically, AdaSVD proposes to update decomposed model weights to reducing reconstruction error and adopt adaptive compression ratio for different layers. Extensive evaluations and analysis have been carried out to demonstrate the performance improvement.

**Strengths:**

1. This paper is well-written and easy to follow. The delicate illustration and text can significantly help readers to better understand the idea of this paper.

2. Experiments is extensive and comprehensive. The experiments covers multiple LLMs from different LLM family as well as different downstream tasks.

3. This paper presents in-depth analysis regarding the proposed method and overall assessment is good, providing a convincing evidence to demonstrate the superiority of the proposed method. However, there are few important concerns needed to be addrees, see Weaknesses belows.

**Weaknesses:**

1. Lack of novelity and seemly incremental contribution. The *AdaComp* is almost the same as the early version of SVD-LLM (https://arxiv.org/pdf/2403.07378v1), where it also adopts a closed-form update to the decomposed matrix. Additionally, the compression ratio allocation strategy in *AdaCR* is not a new thing. This naive compression ratio allocation strategy appears in many submissions to preivous conferences. It originally comes from *Outlier Weighed Layerwise Sparsity (OWL):  A Missing Secret Sauce for Pruning LLMs to High Sparsity*, and the lack of presence of this paper suggests authors seems ignore this important work in their dicussion.

2. Unsound solution. The earlier version of SVD-LLM suffers the critism of concerns against *closed-form update* in their submission to NeurIPS'24, which is *AdaComp* in this work. Essentially, it is kind of cheating since the solution is only *optimal* for this specific data. It can certainly get higher performance on the test set but can lose the generalizability. As for *AdaCR*, it is quite naive and not that convincing. Authors should explore more effective compression ratio allocation stratgies.

**Questions:**

See the weaknesses above.

---

### Official Review · Reviewer_9riQ · 2025-10-30

**Soundness:** 2
**Presentation:** 2
**Contribution:** 2
**Rating:** 2
**Confidence:** 4

**Summary:**

This paper introduces “adaComp” and “adaCR”, two methods designed to enhance SVD-based compression for large language models (LLMs). adaComp iteratively optimizes the decomposed U and V matrices using calibration data, while adaCR adaptively determines the optimal compression ratio for each weight matrix under a given global compression target.

**Strengths:**

1.	The proposed adaComp makes the optimization process more stable.
2.	The proposed method is evaluated on both LLM and VLM, which shows its generality.

**Weaknesses:**

1.	Lack of evaluation on modern LLMs. All experimental results are based on older models such as LLaMA2-7B and Mistral-7B. The study should include evaluations on more recent models, such as the LLaMA3 and Qwen series, to strengthen its relevance and generalizability.
2.	The authors claim that their method outperforms previous approaches across compression ratios ranging from 40% to 80%. However, the presented results only cover the range from 40% to 60%, leaving the higher ratios unverified.
3.	The paper does not compare its approach with several recent and relevant SVD-based compression methods, such as:
[1] SVD-LLM V2: Optimizing Singular Value Truncation for Large Language Model Compression, NAACL 2025
[2] Basis Sharing: Cross-layer Parameter Sharing for Large Language Model Compression, ICLR 2025
Moreover, the relationship between the proposed method and these works is not clearly discussed.
4.	SVD-LLM V2 also determines the optimal compression ratio for each weight matrix, which shares a similar idear with adaCR.

**Questions:**

1.	SVD-LLM find the optimum solution by solving the equation (4) directly. After getting the calibration data, SVD-LLM can compute the solution directly, why there are iteration steps in Fig.3 (a)?
2.	How long does SVD-LLM take for the matrix decomposition and how about yours?
3. The paper does not compare its approach with several recent and relevant SVD-based compression methods, such as:
[1] SVD-LLM V2: Optimizing Singular Value Truncation for Large Language Model Compression, NAACL 2025
[2] Basis Sharing: Cross-layer Parameter Sharing for Large Language Model Compression, ICLR 2025
Moreover, the relationship between the proposed method and these works is not clearly discussed. Please discuss and compare.

---

### Official Review · Reviewer_EiLN · 2025-10-31

**Soundness:** 3
**Presentation:** 2
**Contribution:** 2
**Rating:** 4
**Confidence:** 4

**Summary:**

This paper proposes an adaptive SVD-based compression method for LLMs with two components: (i) adaComp, which alternately refines U/V matrices after SVD truncation using calibration data; and (ii) adaCR, which determines compression ratios by estimating each layer’s importance from similarity-based metrics. Experiments on 7B-scale models (LLaMA-2-7B, OPT-6.7B, Vicuna-7B, Mistral-7B) and LLaVA show that AdaSVD outperforms existing SVD-based methods, including vanilla SVD, FWSVD, ASVD, and SVD-LLM (v1). The method requires calibration data and whitening, and reports perplexity and accuracy across different compression ratios.

**Strengths:**

- Clearly presents two practical improvements to SVD: post-truncation adjustment (adaComp) and layer-wise ratio allocation (adaCR). The SoB description is straightforward and easy to understand.
- Provides a thorough comparison with several SVD baselines (SVD/FWSVD/ASVD/SVD-LLM (v1)) across multiple ratios and LM/VLM benchmarks.
- Implementation details including whitening and a 256-sample calibration set are provided.

**Weaknesses:**

- Limited novelty. Both components mostly build on existing ideas: (a) updating low-rank factors using calibration data is a standard post-truncation method, and (b) non-uniform, layer-wise rank allocation has been studied before (e.g., SVD-LLM, SVD-LLM (v2)). The “importance” metric is just a simple similarity measure without deeper theoretical justification.
- Narrow scope of contribution. The paper focuses on improving rank allocation and post-truncation tuning within the SVD pipeline, rather than proposing a new compression framework. The SoB technique is mainly an engineering workaround for GPU memory limits rather than a scientific contribution.
- Dependence on calibration data. The core update (adaComp) relies on calibration samples, which limits the method’s generality.
- Limited experimental strength. (i) Most of the models used are relatively small and outdated (7B/13B LLaMA-2, OPT-6.7B, Vicuna-7B, Mistral-7B), with no results for newer or larger models like LLaMA-3/3.1/3.2, Qwen2.5, Mistral-Large, or 70B-scale models. (ii) Some important baselines, like SVD-LLM(v2) (NAACL’25) and Dobi-SVD (ICLR’25), are missing, which weakens the claim of achieving SOTA performance.
- Presentation issues. Some figures (e.g., Figure 3) have unclear legends, and tables are sometimes far from the text, which hinders comprehension.

**Questions:**

1. Is the main goal of the paper rank allocation or post-truncation compensation? AdaCR and adaComp seem weakly connected. Could either component work independently as a general method?
2.  Please evaluate more recent and larger models, such as LLaMA-3/3.1/70B, Qwen2.5, and Mistral-Large. Additionally, relevant SOTA baselines like SVD-LLM v2 and ResSVD/Dobi-SVD should be included to provide a fair and up-to-date comparison.
3. How well does the similarity-based metric (Eq. 17–21) indicate which layers are critical, compared with SVD-LLM (v1)’s whitening metric and SVD-LLM v2’s loss-driven allocation? Providing a quantitative correlation analysis would help demonstrate its true effectiveness.
4. Can the proposed procedure be applied to models compressed using non-SVD methods, or even to uncompressed models, as a general low-rank adaptation technique? If not, please clarify its scope.
5. The experimental results of SVD-LLM seem to differ from the original SVD-LLM paper, showing more severe performance drops. Could you clarify any implementation differences?

---

### Official Review · Reviewer_j7PC · 2025-11-04

**Soundness:** 2
**Presentation:** 3
**Contribution:** 2
**Rating:** 4
**Confidence:** 4

**Summary:**

This paper introduces an enhanced SVD-based compression algorithm for large language models (LLMs). The proposed method consists of two key components: (1) iterative optimization of the left and right eigen matrices, and (2) layer-wise compression guided by importance scores. Empirical experimental results demonstrate the effectiveness of the approach, showing competitive performance in terms of perplexity and language understanding benchmarks.

**Strengths:**

1.	The proposed method achieves state-of-the-art performance among SVD-based compression techniques across multiple model architectures and evaluation metrics.

2.	It demonstrates practical engineering contributions, including the introduction of a stack-of-batch technique that improves memory manipulation efficiency during compression and inference.

**Weaknesses:**

1.	The methodological novelty appears limited, offering relatively few new insights to the field. The centered compression proximal objective (Eq. 5) is already well-established in prior literature and has a known closed-form optimal solution. Empirically solving it through iterative optimization lacks theoretical justification, and it remains unclear why a suboptimal solution yields better empirical performance. Additionally, the layer-wise compression using importance scores has been explored in several existing works.

	2.	The paper would benefit from additional experiments and analyses, such as comparisons with other structured pruning methods (with and without fine-tuning), as well as evaluations of compression efficiency and inference latency.

**Questions:**

Could you add additional experiments to address weakness 2?
Also, could you explain why a suboptimal solution to Eq. 5 tends to perform better in practice, rather than only in certain special cases?

---

### Note · Authors · 2025-11-12

I have read and agree with the venue's withdrawal policy on behalf of myself and my co-authors.